# The surprising efficiency of temporal difference learning for rare event prediction

**Xiaoou Cheng**
Courant Institute of Mathematical Sciences
New York University
New York, NY 10012
`chengxo@nyu.edu`

**Jonathan Weare**
Courant Institute of Mathematical Sciences
New York University
New York, NY 10012
`weare@nyu.edu`

## Abstract

We quantify the efficiency of temporal difference (TD) learning over the direct, or Monte Carlo (MC), estimator for policy evaluation in reinforcement learning, with an emphasis on estimation of quantities related to rare events. Policy evaluation is complicated in the rare event setting by the long timescale of the event and by the need for *relative accuracy* in estimates of very small values. Specifically, we focus on least-squares TD (LSTD) prediction for finite state Markov chains, and show that LSTD can achieve relative accuracy far more efficiently than MC. We prove a central limit theorem for the LSTD estimator and upper bound the relative asymptotic variance by simple quantities characterizing the connectivity of states relative to the transition probabilities between them. Using this bound, we show that, even when both the timescale of the rare event and the relative error of the MC estimator are exponentially large in the number of states, LSTD maintains a fixed level of relative accuracy with a total number of observed transitions of the Markov chain that is only *polynomially* large in the number of states.

## 1 Introduction

Prediction of the future behavior of a dynamical system from time series observations is a fundamental task in science and engineering. One need only remember the weather forecast consulted this morning to appreciate the substantial intersection this problem has with our everyday lives. Beyond its own obvious importance, prediction is also a key component in modern machine learning tasks like reinforcement learning, where the rules governing the dynamical system are adjusted to optimize some reward. Exciting applications in that field, like self-driving cars, have captured public imagination.

Prediction is made particularly difficult when the future behavior of interest involves some rare or extreme event. Examples include extreme weather or climate events, failure of reliable engineering products, and the conformational rearrangements that determine critical functions of biomolecules in our bodies. Precisely because of their out-sized impact on our lives and society, rare and extreme events are the most important to predict. Unfortunately, by definition, data sets often contain relatively few examples of these events. In this article we ask: *can rare events be predicted accurately with time series data sets much shorter than the typical timescale of the event?* The surprising answer is Yes. Our mathematical results support recent studies demonstrating accurate rare event prediction with limited data using methods related to those studied here, including in state-of-the-art application settings and for very rare events [Thiede et al., 2019, Strahan et al., 2021, Finkel et al., 2021, 2023b, Antoszewski et al., 2021, Lucente et al., 2021, Finkel et al., 2023a, Strahan et al., 2023a, Jacques-Dumas et al., 2023, Strahan et al., 2023b, Guo et al., 2024].

Within the context of reinforcement learning, "prediction" refers to policy evaluation, that is, calculation of the value function of states given a policy [Sutton and Barto, 2018]. Prediction is the

foundation for policy improvement in control problems. In this work, we focus on prediction for Markov chains on a finite state space. While many dynamical systems of physical interest evolve in continuous spaces, this so-called "tabular" setting is the most common for mathematical analysis. Moreover, some practical algorithms for prediction problems in continuous space begin with a projection onto a finite state Markov process [Thiede et al., 2019, Finkel et al., 2021, 2023a, Jacques-Dumas et al., 2023]. For those algorithms our analysis can be viewed as addressing estimation (but not approximation) error .

Specifically, we formulate a prediction problem in terms of a Markov reward process (MRP) $([n], P, R)$, where $[n] = \{1, 2, \ldots, n\}$ is the state space, $P$ is the Markov transition kernel with $P(i, j) = \mathbf{P}\left[X_{t+1} = j \mid X_t = i\right]$, and $R$ is a deterministic, non-negative reward function. Time dependence and randomness of $R$ can be included by an enlargement of the state space. Our goal is to estimate a value function of the general form[1]

$$u(i) = \mathbf{E}_i \left[ \sum_{t=0}^{T} R(X_t) \right], \tag{1}$$

where $D$ is a subset of $[n]$, $T = \min\{t \geq 0 : X_t \notin D\}$ is the escape time from $D$, and the subscript $i$ on the expectation indicates that $X_0 = i$. By appropriate choice of $R$, we can consider both cases in which the escape time from $D$ is very large and cases in which only low probability escape paths accumulate significant reward. These are the typical situations for rare event statistics, where the reward $R$ and the set $D$ are specified by the application. In the example of self-driving cars, when investigating the rare event that the car crashes, $D$ is the set of all states in which the car has not crashed.

## 1.1 Monte Carlo and temporal difference learning

The prediction problem has been extensively analyzed in the reinforcement learning literature [Sutton and Barto, 2018, Dann et al., 2014]. Our problem's main point of departure from that literature is the central role of the escape event from $D$ in (1). We are specifically interested in cases in which $\mathbf{E}_j[T]$ can be very large so that the length of a single escape trajectory can be very large, and/or cases in which large values of the reward functions are very unlikely so that $u$ can be very small.

Our task is to learn an estimate of $u$ from a set of $M$ samples $\{X_0^\ell, X_1^\ell, \ldots, X_{\tau \wedge T^\ell}^\ell\}_{\ell=1}^M$ of $X_0, X_1, \ldots, X_{\tau \wedge T}$ with $\{X_0^\ell\}_{\ell=1}^M$ drawn from some (possibly unknown) initial probability distribution $\mu$ which is supported in $D$. Here and below we use the shorthand $a \wedge b = \min\{a, b\}$. The deterministic positive integer $\tau$, which we will refer to as a lag time, limits the length of the trajectories in the data set. In practical applications, the sample trajectories are often correlated, for example because multiple length $\tau$ trajectories can be harvested from a single longer trajectory. Here, to simplify the analysis, we assume that the trajectories are drawn independently.

The most direct estimator of $u(i)$ is the Monte Carlo (MC) estimator

$$\hat{u}(i) = \frac{1}{M_i} \sum_{\ell=1}^{M_i} \sum_{t=0}^{T^\ell} R\left(X_t^\ell\right), \quad \text{when } M_i > 0,$$

where $M_i$ is the number of samples with $X_0^\ell = i$, i.e. $M_i = \sum_{\ell=1}^M \mathbf{1}_{\{i\}}(X_0^\ell)$. The estimator is undefined for $i \in D$ with $M_i = 0$. The MC estimator does not allow finite values of $\tau$ (unless $T$ is bounded by a constant). When the escape event is rare, $T$ is very large and the MC estimator requires data sets containing very many observed time steps of $X_t$. Moreover, estimators of rare event statistics are evaluated based on the amount of data required to achieve a desired *relative error* [Bucklew, 2004]. In the rare event setting, $u(i)$ can be extremely small because of the low probability of the event to happen. In this case, assuming $\mu(i) > 0$, control of the relative variance, $\mathbf{Var}\left[\hat{u}(i)\right] / u^2(i)$, often requires $M \propto 1/u(i)$ trajectory samples for the MC estimator. This scaling is fatal when $u(i)$ is very small.

Temporal difference (TD) schemes, on the other hand, express $u$ as the solution to a certain Bellman equation [Sutton, 1988]. They enforce temporal local consistency of the estimate of the value function.

---

[1]General MRPs include a discounting factor $\gamma \in (0, 1]$. We choose $\gamma = 1$ and assume that $u$ is well-defined under this condition. In general, smaller values of $\gamma$ make the prediction problem easier.

As such, TD methods can accommodate *any* positive choice of $\tau$, while MC methods require that each trajectory be observed until escape from $D$ (i.e. $\tau = \infty$). TD schemes are used heavily in reinforcement learning and they are often observed to out-perform MC [Sutton and Barto, 2018]. However, a quantitative understanding of the advantages of TD compared to MC remains elusive. In practice, there are several variants of TD schemes: parametric or tabular, online or batch. In this work, we stick to the simplest setting: the least-squares TD (LSTD) estimator [Bradtke and Barto, 1996] in the tabular setting of a finite state MRP. Our goal is to clearly characterize the benefit of the LSTD estimator over MC for predictions related to rare events in this setting.

TD schemes begin with the observation that, for any choice of $\tau$, $u$ in (1) solves the linear system

$$(I - S^\tau)u(i) = \sum_{t=0}^{\tau-1} S^t R_D(i) \text{ for } i \in D \quad \text{and} \quad u(i) = R(i) \text{ for } i \notin D \tag{2}$$

where

$$S(i,j) = P(i,j) \text{ for } i \in D \quad \text{and} \quad S(i,j) = \delta_{ij} \text{ for } i \notin D,$$

is the transition operator of $X_{t \wedge T}$, $S^\tau$ is the $\tau$th power of $S$ and $R_D(i) = R(i)$ for $i \in D$ while $R_D(i) = 0$ for $i \notin D$. In the limit $\tau \to \infty$, (2) becomes (1). In our theory and examples, we will see that the choice of $\tau$ can have a dramatic effect on the performance of the estimator.

TD methods use trajectory data to find an approximate solution to (2). Specifically, the LSTD estimator $\tilde{u}$ in this tabular setting is the solution to a linear system where the transition matrix $S$ in (2) is approximated with trajectory data. We set $\tilde{S}^0 = I$ and, for $t = 1, 2, \ldots, \tau$,

$$\tilde{S}^t(i,j) = \frac{1}{M_i} \sum_{\ell=1}^M \mathbf{1}_{\{(i,j)\}} \left( X_0^\ell, X_{t \wedge T^\ell}^\ell \right) \text{ for } M_i > 0 \quad \text{and} \quad \tilde{S}^t(i,j) = \delta_{ij} \text{ for } M_i = 0.$$

Unlike $S^t$, $\tilde{S}^t$ are not powers of a single matrix. When it exists, the $\tau$-step LSTD estimator, $\tilde{u}$ solves

$$(I - \tilde{S}^\tau)\tilde{u}(i) = \sum_{t=0}^{\tau-1} \tilde{S}^t R_D(i) \text{ for } i \in D \quad \text{and} \quad \tilde{u}(i) = R(i) \text{ for } i \notin D. \tag{3}$$

(Like the MC estimator, $\tilde{u}(i)$ is undefined when $M_i = 0$). In the $\tau \to \infty$ limit, the $\tau$-step LSTD and MC estimators are equivalent. We therefore occasionally refer to the Monte Carlo estimator as the $\tau = \infty$ estimator.

## 1.2 Our contributions

We pause now to consider what classical perturbation theory for linear systems has to say about the feasibility of estimating $u$, the solution of (2), by $\tilde{u}$, the solution to (3). Clearly (3) only requires access to trajectories of length $\tau$, but how sensitive is the solution of (2) to perturbations in $S$? If it is very sensitive then any reduction in data requirements from shorter trajectories may be offset by a need for many more trajectories. Classical perturbation theory tells us that errors $S^t - \tilde{S}^t$ can, in the worst case, be amplified in $u - \tilde{u}$ by a factor of the condition number $\kappa^\tau = \|(I - S_D^\tau)^{-1}\| \|I - S_D^\tau\|$, where $S_D^\tau$ is the matrix obtained from $S^\tau$ by setting to zero any row or column with index in $D^c$ [Demmel, 1997]. As the next lemma shows, when typical values of $T$ are much larger than $\tau$, $\|(I - S_D^\tau)^{-1}\|$ will be very large. The proof of Lemma 1 is in Appendix A.

**Lemma 1.** *For any consistent matrix norm, if the restriction of $S_D$ to row and column indices in $D$ is irreducible and aperiodic, then $\|(I - S_D^\tau)^{-1}\| \geq \mathbf{E}_\nu[T]/\tau$ where $\nu(i) = \lim_{t \to \infty} \mathbf{P}[X_t = i \mid T > t]$ is the quasi-stationary distribution and the subscript on the expectation indicates that $X_0$ is drawn from $\nu$. (See Collett et al. [2012] for more about the quasi-stationary distribution).*

While it is possible that a small value of $\|I - S_D^\tau\|$ can compensate for a large value of $\mathbf{E}_\nu[T]/\tau$ and result in a moderate condition number, for many rare event problems this is not the case. For example, for the system studied in our numerical experiments in Section 3, $\|I - S_D\|$ is bounded away from zero even as the escape event becomes increasingly rare (see Appendix B).

Thus it would seem that we are forced to choose between two doomed options: choose $\tau$ small to control the length of trajectories we need to observe but observe a huge number ($M \propto \mathbf{E}_\nu[T]/\tau$) of

them to drive down the error in $\tilde{S}^t$, or observe very long trajectories ($\tau \propto \mathbf{E}_\nu[T]$) to control $\kappa^\tau$. With either of these choices we would not expect TD to significantly outperform MC.

A primary goal of this article is to explain that, in many (perhaps most) cases, this is a false choice. The worst case analysis that characterizes the classical perturbation theory is, by design, pessimistic. But as we will explain, in our setting it is wildly pessimistic. With a few simple and practically relevant assumptions we will be able to show that $\tau$-step LSTD can achieve remarkably accurate estimates with remarkably little data.

Throughout this paper we quantify the relative accuracy of $\tilde{u}(i)$ by the *relative asymptotic variance* $\sigma_i^2/u^2(i)$ where $\sigma_i^2$ is the variance of the limiting normal distribution in a central limit theorem for $\tilde{u}(i)$ established in Theorem 1. Also in Theorem 1, we provide an upper bound on the relative asymptotic variance by simple quantities characterizing the connectivity of states relative to the transition probabilities between them. Crucially, neither the condition number $\kappa^\tau$, nor any expectation of $T$, appear explicitly in the bound. Next, we turn our focus to the rare event setting, which we characterize via the large $n$ behavior of the estimator. In this setting both the typical escape time and the relative variance of the MC estimator can scale exponentially with $n$. As we show in Theorem 2, however, the relative asymptotic variance of the LSTD estimator scales, at worst like $n^3$.

## 1.3 Related work

Since it first appeared in [Sutton, 1988], there have been numerous variants of TD [Bradtke and Barto, 1996, Dann et al., 2014, Lillicrap et al., 2019]. Early theoretical analysis of TD focused on asymptotic convergence with linear value function approximation [Jaakkola et al., 1993, Dayan and Sejnowski, 1994, Tsitsiklis and Van Roy, 1997], along with examples of divergence [Baird, 1995, Tsitsiklis and Van Roy, 1997]. More recently, nonasymptotic convergence analysis of TD has been performed [Bhandari et al., 2018, Dalal et al., 2018, Srikant and Ying, 2019, Cai et al., 2019].

Besides its own convergence, a long-considered question for TD is when and how it outperforms MC. While practical experience suggests that TD is more efficient than MC, a comprehensive theory is lacking. Grunewalder et al. [2007] and Grünewälder and Obermayer [2009] proved that LSTD is at least as statistically efficient as MC, but the improvement is not quantified. Cheikhi and Russo [2023] expresses the relative benefit of TD over MC by the inverse trajectory pooling coefficient, which facilitates interpretation of the ratio of asymptotic variances. But how that translates to quantitative improvements requires further elucidation. As far as we are aware, prediction for rare event problems has not been analyzed in the reinforcement learning literature.

TD enforces temporal consistency for trajectory data. Trajectory data has been utilized in applications more broadly. Markov state models [Husic and Pande, 2018] and dynamic mode decomposition methods [Schmid, 2022] were developed in the molecular dynamics and fluid dynamics communities respectively and have been successful at estimating eigenvectors and eigenvalues of Markov transition (or Koopman) operators from trajectory data in state-of-the-art applications. The dynamical Galerkin approximation [Thiede et al., 2019] method extends those approaches to the prediction problem studied here and has been used to study rare events in molecular dynamics and climate science [Strahan et al., 2021, Antoszewski et al., 2021, Finkel et al., 2021, 2023a, Guo et al., 2024]. Despite widespread application in the study of various rare events over several decades, the present article is the first theoretical evidence that trajectory analysis methods can be effective tools specifically for rare events.

## 1.4 Notation

$X \xrightarrow{\mathcal{D}} \mathcal{N}(\mu_0, \sigma_0^2)$ indicates that a random variable $X$ converges in distribution to a normal distribution with mean $\mu_0$ and variance $\sigma_0^2$. For any subset $A \subseteq [n]$, we define the hitting time $T_A = \min\{t > 0 : X_t \in A\}$. When $X_t$ never hits $A$, $T_A = +\infty$. When the set $A$ contains only one state $i$, we use $T_i$ for simplicity. Note that in (1), $T$ counts from $t = 0$ instead. While $S$ is the transition probability matrix for the Markov chain $X_{t \wedge T}$, we define another Markov chain $Y_t^\tau$ whose transition probability matrix is $S^\tau$. For $Y_t^\tau$, we define its hitting time to be $T_A^\tau = \min\{t > 0 : Y_t^\tau \in A\}$. We define $T^\tau = \min\{t \geq 0 : Y_t^\tau \in D\}$ as a counterpart of $T$ for $X_t$. We use $e_i$ for the $i$th standard basis vector, i.e. $e_i(j) = \delta_{ij}$. For any two quantities $a$ and $b$, $a \gtrsim b$ means that there exists a positive constant $C$ independent of $n$ such that $a \geq Cb$. Similarly, $a \lesssim b$ means that $a \leq Cb$.

## 2 An upper bound for relative asymptotic variance

As previous authors have done [Cheikhi and Russo, 2023], we will characterize the error of the LSTD estimator using a central limit theorem. Unlike previous work, we will also provide a simple upper bound on the relative asymptotic variance that clearly distinguishes the LSTD estimator from the MC estimator. Our subsequent analysis of the LSTD estimator in the rare event setting will be derived from this bound.

**Theorem 1.** *Suppose $\mu(i) > 0$ for all $i \in D$ and $P$ is irreducible. Then, as the number of samples $M \to \infty$,*

$$\sqrt{M}\left(\tilde{u}(i) - u(i)\right) \xrightarrow{\mathcal{D}} \mathcal{N}(0, \sigma_i^2), \tag{4}$$

*with*

$$\sigma_i^2 = \sum_{k \in D} \frac{1}{\mu(k)} \left(e_i^\mathsf{T}(I - S_D^\tau)^{-1}e_k\right)^2 \mathbf{E}_k \left[\sum_{t=0}^{\tau-1} R_D(X_{t \wedge T}) + u(X_{\tau \wedge T}) - u(k)\right]^2. \tag{5}$$

*Moreover, the relative asymptotic variance $\sigma_i^2/u^2(i)$ satisfies the upper bound*

$$\frac{\sigma_i^2}{u^2(i)} \leq \sum_{\substack{k \in D,\, \ell \in [n], \\ \ell \neq k}} \frac{\sum_{t=1}^{\tau} S^t(k, \ell)}{\mu(k)\, Q^\tau(k, \ell)^2} \tag{6}$$

*where we have introduced the key quantity $Q^\tau(k, \ell) = \mathbf{P}_k\left[T_\ell^\tau < T_k^\tau \wedge T_{D^c \setminus \{\ell\}}^\tau\right]$.*

An important special case that will include one of the examples studied in Section 3, occurs when the reward function $R$ is zero in $D$ (but non-zero in $D^c$). In this case we can strengthen the bound in Theorem 1 by only including $S^\tau$ in the numerator.

**Corollary 1.** *Under the same assumptions as in Theorem 1 but with $R(i) = 0$ for any $i \in D$,*

$$\frac{\sigma_i^2}{u^2(i)} \leq \sum_{\substack{k \in D,\, \ell \in [n], \\ \ell \neq k}} \frac{S^\tau(k, \ell)}{\mu(k)\, Q^\tau(k, \ell)^2}. \tag{7}$$

Let us take a moment to parse the bounds in Theorem 1 and Corollary 1. The bounds tell us that if the sum is bounded, our estimator achieves entry-wise *relative* accuracy, indicating that even when $u(i)$ is extremely small, the error in $\tilde{u}(i)$ can be much smaller (depending only on $M$). But the denominator is the real star of the show. It is the sum of the probabilities of all paths connecting state $k$ and state $\ell$ that do not return to $k$ and do not exit $D$ (except possibly through $\ell$). In particular $Q^\tau(k, \ell) \geq S^\tau(k, \ell)$. While the probability $Q^\tau(k, \ell)$ can be very small for most pairs $(k, \ell)$, evidently it is the size of $Q^\tau(k, \ell)^2$ relative to $S^t(k, \ell)$ for $t \leq \tau$ (or only $S^\tau(k, \ell)$) that matters. Fortunately, in many cases, we can expect $Q^\tau(k, \ell)^2$ to be *much* larger than $S^\tau(k, \ell)$ for a good choice of $\tau$. We will quantify this statement in the rare event setting in Section 4.

A quantity very similar to $Q^\tau(k, \ell)$ appears in the matrix analysis literature [Thiede et al., 2015] as a measure of the sensitivity of the relative accuracy of the invariant distribution of a Markov chain to entry-wise perturbations in its transition probability matrix. The proofs of Theorem 1 and Corollary 1 are given in Appendix C. Lemma 2 in Appendix C, which is used to bound the variation in $u$ between states, is key to the argument.

## 3 Experiments

We now consider two example problems that illustrate the challenges posed by rare event prediction. For both problems the statistical efficiency of the MC estimator degrades exponentially in the state space size $n$. Our goal is to see whether or not LSTD's performance degrades similarly as $n$ increases, and, in so doing, to motivate our final bounds on the relative asymptotic variance of LSTD established later in Section 4. The examples and numerical results presented in this section are exactly consistent with the assumptions and results presented in Section 4.

Both problems are based on a one dimensional nearest neighbor chain $X_t \in [n]$ with transition rules

$$P(i, i \pm 1) = \frac{p(i \pm 1)}{2(p(i) + p(i \pm 1))} \text{ and } P(i, i) = 1 - P(i, i+1) - P(i, i-1) \text{ for } i \pm 1 \in [n] \tag{8}$$

with $P(1,0) = P(n, n+1) = 0$ on the boundary. Here $p$ is the invariant probability vector of $P$,

$$p(i) \propto \exp\left[\frac{n-1}{4\pi} \cos\left(\frac{4\pi(i-1)}{n-1}\right)\right], \tag{9}$$

which has modes around $i = 1$, $i = (n+1)/2$, and $i = n$, that become more sharply peaked as $n$ increases. The escape time from any of these modes scales exponentially with $n$. We will be interested in predictions related to the escape of $X_t$ from $D = [n] \setminus \{1, n\}$, and choose $\mu$ to be the uniform distribution on $D$.

Before moving on to the specific problem statements, we point out that despite the exponentially long waiting time to transition between the modes of $p$, transitions between nearest neighbor states remain stable as $n$ increases. In fact, for $i \in D$, $P(i, i \pm 1) \geq \frac{1}{2(1+e)}$. Stable local transition probabilities such as these play a key role in our upper bounds in Section 4, where they are used to lower bound $Q(k, \ell)$ in (6).

**The mean first passage time.** The mean first passage time $u(i) = \mathbf{E}_i[T]$ solves (2) with $R(i) = 1$ for $i \in D$ and $R(i) = 0$ for $i \notin D$. We plot the mean first passage time for $n = 20,\ 40$, and 80 in the left panel of Fig. 1. Evidently the largest values of $\mathbf{E}_i[T]$ scale exponentially with $n$. In Appendix B, we prove that, near $i = (n+1)/2$,

$$\mathbf{E}_i[T] \gtrsim \exp\left(\frac{3n}{8\pi}\right). \tag{10}$$

Therefore the trajectories required by the MC estimator will include a number of transitions of $X_t$ that scales exponentially with $n$. We compare the performance of the MC estimator of the mean first passage time to that of the LSTD estimator, $\tilde{u}$, found by solving (3) with $\tau = 1$. In the middle panel of Fig. 1 we plot the relative asymptotic variance, $\sigma_i^2/u^2(i)$, for the same values of $n$. In the same panel we plot empirical estimates of the the (non-asymptotic) relative mean squared error (MSE), $M\mathbf{MSE}\left[\tilde{u}(i)\right]/u^2(i)$, with $M = 10n^3$. The empirical relative MSE estimates are computed by generating 30000 independent copies of the LSTD estimator. Remarkably, the relative MSE of the LSTD estimator grows *more slowly* than $n^3$. Meanwhile, near the boundary of $D$, the relative MSE of the MC estimator (computed exactly) grows exponentially fast with $n$, as can be seen in the right panel of Fig. 1.

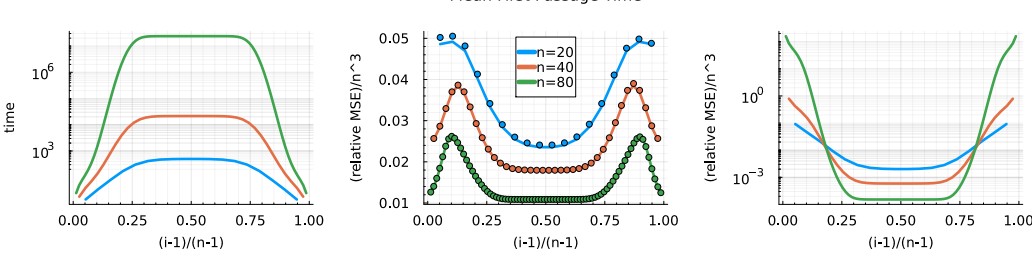

Figure 1: Left: the (exact) mean first passage time $u(i)$ with $n = 20,\ 40$, and 80. Middle: the relative asymptotic variance (solid lines) and the relative empirical MSE (circles) of the LSTD estimator with $\tau = 1$. The relative empirical MSE are obtained with sample sizes $M = 10n^3$. Right: the (exact) relative asymptotic MSE of the MC estimator.

**The committor.** We now consider estimation of the committor function

$$u(i) = \mathbf{P}_i\left[T_n < T_1\right] \text{ for } i \in [n] \setminus \{1, n\} \quad \text{and} \quad u(i) = \mathbf{1}_{\{n\}}(i) \text{ for } i \in \{1, n\}.$$

The committor corresponds to the choice $R(i) = 0$ for $i \neq n$ and $R(n) = 1$. Because $\mathbf{E}_i[T]$ can be exponentially large, the MC estimator of the committor again requires a data set containing exponentially long trajectories of $X_t$. Moreover, in Appendix B, we prove that $u(i)$ can be exponentially small in $n$, as

$$u(2) \lesssim \exp\left(-\frac{n}{4\pi}\right), \tag{11}$$

as is evident in the plot of the committor for $n = 20,\ 40$, and 80, in the left panel of Fig. 2. As a consequence, the relative MSE of the MC estimator, plotted in the right panel of Fig. 2, grows

exponentially fast with $n$. As can be seen from the middle panel of Fig. 2, however, the relative asymptotic variance of the LSTD committor estimator with $\tau = 1$ grows *more slowly* than $n^3$. Again empirical estimates of the relative MSE of the LSTD estimator with $M = 10n^3$ agree well with corresponding relative asymptotic variances and show the same trend.

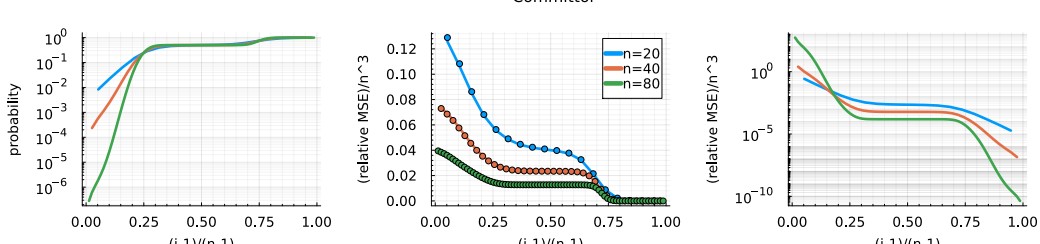

Figure 2: Left: the (exact) committor function $u(i)$ with $n = 20, 40$, and 80. Middle: the relative asymptotic variance (solid lines) and the relative empirical MSE (circles) of the LSTD estimator with $\tau = 1$. The relative empirical MSE are obtained with sample sizes $M = 10n^3$. Right: the (exact) relative asymptotic MSE of the MC estimator.

Together, the mean first passage time and the committor estimation problems in this section strongly suggest a significant (polynomial versus exponential in $n$) advantage for TD methods over MC. In the next section we will prove that the performance advantage observed on these two examples holds in significant generality.

**Effect of the lag time.** In the above experiments we have examined the error of the LSTD estimator with $\tau = 1$. In practice, the best choice of $\tau$ is long enough to avoid a nearly diagonal $S^\tau$, but not so long as to lose the efficiency advantage of TD over MC that we have just observed. Our bounds in Theorem 1 also change with $\tau$. To better illustrate the effect of $\tau$ on the relative asymptotic variance of the LSTD estimator and on our bounds, we now consider a "lazier" version of the Markov chain in (8). Specifically, we define $P(i, i \pm 1) = \frac{p(i\pm1)}{10(p(i)+p(i\pm1))}$ for $i \pm 1 \in D$, with the same boundary conditions as before. We again consider estimates of the mean first passage time and the committor, now with fixed $n = 40$. We plot the maximum of the relative asymptotic variance, $\sigma_i^2/u^2(i)$, over $i \in D$ in Fig. 3, along with the bound (6) for the mean first passage time in the left panel and (7) for the committor in the right panel. Though the plots do not extend to values of $\tau$ comparable with the largest values of $\mathbf{E}_i[T]$, the relative variance of the MC estimator corresponds to the asymptote of the true relative variance toward the right hand side of each plot. We observe that the true relative variance of the LSTD estimator is minimized in both cases for a choice of $\tau \approx 30$ and that, for this choice, the relative variance of the LSTD estimator is significantly smaller than the relative variance of the MC estimator. The bounds in (6) and (7) are designed to capture the high accuracy of LSTD for relatively small values of $\tau$ and we indeed see that they deteriorate as $\tau$ becomes very large. However, at least in the committor case, the bound accurately reproduces the initial decrease in error that occurs when $\tau$ is increased above $\tau = 1$.

**Effect of the initial distribution.** So far, in our experiments, we have chosen $\mu$ to be the uniform distribution on $D$. A natural alternative strategy is to harvest many short trajectories from a single, much

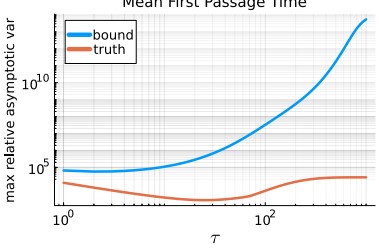 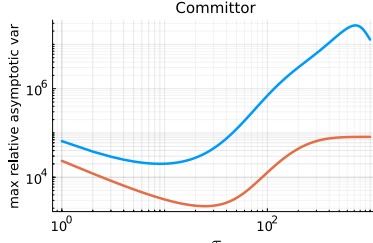

Figure 3: The bound and the truth for the maximum relative asymptotic variance of the mean first passage time and the committor with varying lag time $\tau$. The number of states fixed at $n = 40$. The relative asymptotic variance bounds for the mean first passage time and the committor are from (6) and (7) respectively.

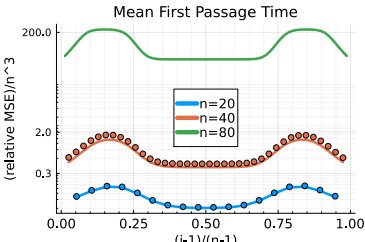 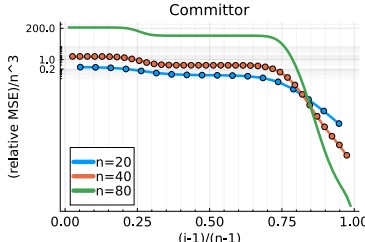

Figure 4: The relative asymptotic variance (solid lines) and the relative empirical MSE (circles) of the LSTD estimators of the mean first passage time and the committor, with $\mu$ being the invariant distribution $p$ conditioned within $D$. Note the scale is logarithmic. The relative empirical MSE are obtained with sample sizes $M = 10n^3$. For $n = 80$ the TD estimator fails with high probability and the empirical error is undefined.

longer trajectory. Under ergodicity assumptions on $P$, these short trajectories are then approximately drawn from the invariant distribution of $P$. In Fig. 4, we present the relative asymptotic variance and the empirical estimates of the relative MSE of the mean first passage time and the committor, with $P$ being the same as in (8), and $\mu$ chosen to be the invariant distribution $p$ in (9) conditioned within $D$. For $\tau = 1$ and $n = 20, 40$, and $80$, the maximum relative asymptotic variances of the LSTD estimators grow exponentially with $n$. With $M = 10n^3$, the empirical relative MSE estimators with 30000 independent copies of the LSTD estimators agree well with the relative asymptotic variance for $n = 20$ and $40$. For $n = 80$, the empirical linear system (3) fails to define the LSTD estimators with high probability, and the empirical error is undefined. These results should be contrasted with the much smaller errors shown in the middle panels of Figs. 1 and 2, corresponding to uniformly chosen initial conditions. We expect that the results in Fig. 4 would be somewhat different if, instead of initializing trajectories independently from the conditional invariant distribution, we had harvested correlated initial conditions from a single ergodic trajectory. Nonetheless, the results indicate that the choice of $\mu$ can have a significant impact on the performance of LSTD for prediction of rare events.

## 4   Rare event assumptions and upper bound

Our final relative asymptotic variance bounds will rigorously establish the dramatic advantage of TD over MC that we observed in the experimental results of Section 3. As in that section, we focus on the behavior of relative variance as $n$ increases. Our results rely on several basic assumptions. On the one hand, all of these assumptions are satisfied by the Markov chain examined in Section 3. On the other hand, the results in this section cover considerably more general Markov chains. The assumptions concern certain structural properties of the Markov chain as $n$ increases, but crucially, they allow both the typical escape time and the relative variance of the MC estimator to grow *exponentially* in $n$.

To ensure that a unique $\tilde{u}$ will exist for large enough $M$, we make the following minorization assumption on $\mu$.

**Assumption 1** (Lower bound on $\mu$). *For some constant $\alpha > 0$, independent of $n$, and all $i \in D$, $\mu(i) \geq \frac{\alpha}{n}$.*

Recall that the uniform distribution on $D$ was used to generate initial conditions for the numerical tests described in Section 3. The invariant distribution $p$ in (9) conditioned within $D$, instead, violates this assumption, as its minimum is exponentially small in $n$.

As mentioned in Section 1.2 the perturbations, $\tilde{S}^\tau - S^\tau$, relevant for analysis of the LSTD estimator are far from the worst case perturbations characterizing classical perturbation bounds for Eq. (2). Each entry of the matrix $\tilde{S}^t$ is the empirical frequency of a specific transition. As a result, the variance of $\tilde{S}^t(i,j)$ is proportional to $S^t(i,j)\,(1 - S^t(i,j)) \leq S^t(i,j)$, and we can characterize the perturbations by characterizing the entries themselves. Different entries of $S^t$ can have very different magnitudes, and many Markov processes exhibit higher transition probabilities between states that are "close" according to some metric and small transition probabilities between states that are "far away". To express this notion in assumptions without resorting to any topology within which the state space may be embedded, we upgrade $[n]$ to an unweighted directed graph $G$ by including edges

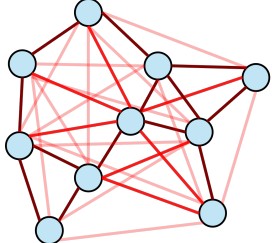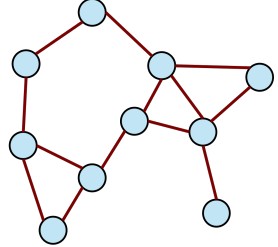

Figure 5: Left: an undirected graph with edges colored according to the transition probabilities. An edge with darker color corresponds to a transition with higher probability. Right: after pruning edges with low transition probabilities, the minorizing graph stays connected.

only along transitions with significant probability. Our additional assumptions will be stated in terms of properties of this "minorizing graph."

Specifically, we introduce a directed edge from state $i \in G$ to another state $j \in G$ if $S^\tau(i,j) \geq c$, where $c$ is a positive constant independent of the number of states $n$.

**Assumption 2** (Connected minorizing graph). *For some constant $c > 0$ independent of $n$, any two states $i \in D$ and $j \in [n]$ are connected by a path in $G$.*

When $P$ is irreducible, as assumed in Theorem 1, and $n$ is fixed, we can always find a $c > 0$ so that any $i \in D$ and $j \in [n]$ are connected by a path in $G$ for $\tau = 1$. Assumption 2 emphasizes the independence of $n$, asserting that the transition probabilities above the threshold $c$ can, on their own, preserve the connectivity of the graph while $n$ increases. Recall that the Markov chain studied in Section 3 satifies $P(i, i \pm 1) \geq 1/(2(1 + e))$, implying that Assumption 2 is satisfied with the choice $c = 1/(2(1 + e))$.

Let $d(i, j)$ be the length of the shortest path from state $i$ to state $j$ in $G$. When the graph is undirected, $d$ is a distance on $G$, but it need not be. Finally, we characterize the decay of the transition probabilities of pairs of states that are "far away" according to $G$.

**Assumption 3** ("Sub-Gaussian" transition probability tails). *For some constants $\beta > 0$ and $C$, independent of $n$, for all $i \neq j$, and $t \leq \tau$, $S^t(i,j) \leq Ce^{-\beta d^2(i,j)}$.*

Assumption 3 stipulates that transition probabilities decay fast enough for distant states in $G$. For the Markov chain studied in Section 3, when $\tau = 1$ this assumption is satisfied because transition probabilities are zero for states that are not nearest neighbors. Assumptions 2 and 3 essentially characterize the locality of the transitions. These assumptions are practically reasonable even when, as in the example in Section 3, the global landscape of the transitions exhibits challenging traits such as multi-modality.

**Illustration.** *The left panel of Fig. 5, depicts a general (symmetric) Markov chain on a graph. Darker edges indicate higher transition probabilities. The transition probabilities are chosen to decay with Euclidean distance in the plane, consistent with, for example, a Gaussian transition kernel in the plane. The minorizing graph on the right retains only those edges exceeding the threshold $c$, which is chosen large enough to remove most distant edges, but small enough to result in a connected graph.*

In Section 3, we considered the effect of the choice of $\tau$ on relative asymptotic variance. In that example, a larger choice of $\tau$ initially increased the probability of nearest neighbor transitions and would allow for a larger choice of $c$. As $\tau$ increases further however, the probabilities of distant transitions would become larger, potentially decreasing the maximum allowed choice of $c$ *and* possibly requiring a smaller choice of $\beta$ and/or a larger choice of $C$ in Assumption 3. The quantitative interplay among $c$, $\beta$ and $C$ are determined by the specific Markov chain under study, but we expect similar behavior for common transition probabilities such as Gaussian transition kernels.

As the experiments in Section 3 clearly demonstrate, these assumptions allow both the typical escape time and the relative variance of the MC estimator to scale exponentially with $n$. The failure of MC in these scenarios does not contradict this paper's theoretical results because the MC estimator does not satisfy Assumption 2. Indeed, the MC estimator corresponds to $\tau = \infty$, for which $S^\tau(i,j)$ can only be non-zero if $i \in D$, $j \in D^c$.

With these assumptions, we can now state our upper bound for the rare event setting, whose proof is very simple but informative. The key idea is that, under these assumptions, $Q^\tau(k, \ell)$ in the denominator is indeed much larger than the transition probabilities $S^t(k, \ell)$ in the numerator.

**Theorem 2.** *Under Assumptions 1, 2 and 3, we have the following asymptotic variance bound:*

$$\frac{\sigma_i^2}{u^2(i)} \leq \frac{C}{\alpha}\tau e^{\frac{(\log c)^2}{\beta}}\, n^3. \tag{12}$$

*When $R(i) = 0$ for all $i \in D$,*

$$\frac{\sigma_i^2}{u^2(i)} \leq \frac{C}{\alpha}e^{\frac{(\log c)^2}{\beta}}\, n^3. \tag{13}$$

*Proof of Theorem 2.* Because $Q^\tau(k, \ell)$ is the sum of the probabilities of all paths connecting states $k$ and state $\ell$ that do not return to $k$ and do not exit $D$ (except through $\ell$), Assumption 2 implies that

$$Q^\tau(k, \ell) \geq c^{d(k,\ell)}.$$

Plugging this bound and the ones in Assumptions 1 and 3 into (6) we find that

$$\frac{\sigma_i^2}{u^2(i)} \leq \frac{C}{\alpha}\,\tau\,n \sum_{\substack{k \in D,\, \ell \in [n], \\ \ell \neq k}} e^{-d(k,\ell)(2\log c + \beta\, d(k,\ell))}.$$

Optimizing the summand over $d(k, \ell)$ gives the bound in (12). The bound in (13) follows from the same argument using (7) instead of (6).  □

Remarkably, the total amount of data (measured in observed transitions of $X_t$) required to achieve a fixed relative accuracy has reduced from as bad as exponential in $n$ for the MC estimator to no worse than $n^3$ for the short trajectory estimator.

## 5   Conclusions and future work

In this article we show that the LSTD method can produce relatively accurate estimators of rare event statistics with a data set of observed Markov chain transitions that is dramatically smaller than would be required by the Monte Carlo estimator. In particular, the LSTD estimator can achieve relative accuracy with a number of observed transitions much smaller than typical timescale of the rare event. This contrasts with the classical worst case perturbation bounds, which predict a large error when the typical timescale of the rare event is large.

Generalization of our basic conclusions beyond the tabular setting is a natural goal for future work. Our proof strategy can be used to establish general, entry-wise perturbation bounds for linear systems of the form in (2). A version of those bounds that applies to Markov processes in continuous spaces would have many interesting consequences, including in the analysis of both approximation and estimation error for TD approaches beyond the tabular setting.

We have not considered online TD approaches [Sutton and Barto, 2018]. In the tabular setting considered here, standard online TD corresponds to a variant of classical Richardson iteration in which the residual of (2) is replaced by an independent realization of the residual of (3) at each iteration. Both deterministic Richardson iteration and online TD will converge very slowly when the typical timescale of the rare event is large. This issue has been explored in Strahan et al. [2023b], where the authors suggest a batch version of subspace iteration for online policy evaluation and demonstrate improved convergence. The data requirements of that scheme should be studied theoretically.

## Acknowledgements

This work was supported by the National Science Foundation through award DMS-2054306. We thank Aaron R. Dinner, John Strahan, and Robert J. Webber for their help exploring early versions of our results. We are grateful to Joan Bruna, Yaqi Duan, Yuehaw Khoo, Yiping Lu, Christopher Musco, and Jonathan Niles-Weed for helpful discussions. This work was supported in part through the NYU IT High Performance Computing resources, services, and staff expertise.

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

## A   Proof of Lemma 1

*Proof.* Irreducibiliy and aperiodicity of $S_D$ implies that $\nu$ is the unique left eigenvector of $S_D$ with largest eigenvalue, $\lambda_{\max}$. Using the fact that the $u(i) = \mathbf{E}_i[T]$ solves (2) with $R(i) = 1$ for $i \in D$ and $R(i) = 0$ for $i \notin D$, a direct calculation yields $\mathbf{E}_\nu[T] = (1 - \lambda_{\max}^\tau)^{-1} \mathbf{E}_\nu[T \wedge \tau]$. Therefore

$$\|(I - S_D^\tau)^{-1}\| \geq (1 - \lambda_{\max}^\tau)^{-1} = \frac{\mathbf{E}_\nu[T]}{\mathbf{E}_\nu[T \wedge \tau]} \geq \frac{\mathbf{E}_\nu[T]}{\tau}$$

and

$$\kappa^\tau = \|(I - S_D^\tau)^{-1}\| \|I - S_D^\tau\| \geq \frac{\mathbf{E}_\nu[T]}{\tau} \|I - S_D^\tau\|.$$

$\square$

## B   Properties of the Markov chain studied in Section 3

*Proof of lower bound of $\|I - S_D\|_2$.* Let $\hat{S}_D$ be the restriction of $S_D$ to row and column indices in $D$, i.e. the $(n - 2) \times (n - 2)$ submatrix in the middle of $S_D$. $\hat{S}_D$ is a substochastic matrix that satisfies detailed balance with respect to $p$ in (9) for those entries in $D$. Therefore, $\hat{S}_D$ is similar to a real symmetric matrix, and all eigenvalues of $\hat{S}_D$ are real. Let $\lambda_{\min}$ be the smallest eigenvalue of $\hat{S}_D$. We have

$$\|I - S_D\|_2 \geq 1 - \lambda_{\min}.$$

We will derive an upper bound of $\lambda_{\min}$ to show that $\|I - S_D\|_2$ is bounded away from 0.

Define an $(n - 2) \times (n - 2)$ diagonal matrix $D_p$ as $D_p(i, i) = p(i + 1)$. Note the index adjustment is due to the size difference of $D_p$ and $p$, and similar adjustment will persist in this proof. By the detailed balance property $\hat{S}_D(i, j)p(i + 1) = \hat{S}_D(j, i)p(j + 1)$ for $i, j = 1, \ldots, n - 2$, the matrix $A = D_p^{\frac{1}{2}} \hat{S}_D D_p^{-\frac{1}{2}}$ is a symmetric matrix, and it has the same eigenvalues as $\hat{S}_D$.

Now we will give an upper bound of $\lambda_{\min}$, the smallest eigenvalue of $A$. By Rayleigh quotient characterization of $\lambda_{\min}$, any length $n - 2$ nonzero vector $v$ satisfies $\lambda_{\min} \leq \frac{v^\mathsf{T} A v}{v^\mathsf{T} v}$. Specifically, we choose $v$ as

$$v(i) = \sqrt{\frac{2}{n - 1}} \sin\left(\frac{i(n - 2)\pi}{n - 1}\right), \quad i = 1, 2, \ldots, n - 2.$$

The motivation is that in the limit $n \to \infty$, $A$ will be related to a (weighted) finite difference matrix for the Laplace operator in one dimension, so we would like to use $v$ which is a unit eigenvector corresponding to the smallest eigenvalue of the Laplace operator.

Our remaining work is to calculate $v^\mathsf{T} A v$ and upper bound it. $A$ is a tridiagonal matrix with $A(i, i) = P(i + 1, i + 1)$, $A(i, i + 1) = 1 / \left[ 2\left( \sqrt{\frac{p_{i+1}}{p_{i+2}}} + \sqrt{\frac{p_{i+2}}{p_{i+1}}} \right) \right]$, $A(i - 1, i) = 1 / \left[ 2\left( \sqrt{\frac{p_i}{p_{i+1}}} + \sqrt{\frac{p_{i+1}}{p_i}} \right) \right]$. By direct calculation, with $\alpha = \frac{\pi}{n-1}$ we have

$$\begin{aligned}
v^\mathsf{T} A v = {} & \frac{2}{n - 1} \sum_{i=1}^{n-2} \left[ \left( 1 - \frac{p_{i+2}}{2(p_{i+1} + p_{i+2})} - \frac{p_i}{2(p_{i+1} + p_i)} \right) \frac{1 - \cos(2i\alpha)}{2} \right] \\
& + \frac{2}{n - 1} \sum_{i=1}^{n-2} \frac{1}{4} \frac{1}{\sqrt{\frac{p_{i+2}}{p_{i+1}}} + \sqrt{\frac{p_{i+1}}{p_{i+2}}}} \left[ -\cos\alpha + \cos((2i + 1)\alpha) \right] \\
& + \frac{2}{n - 1} \sum_{i=1}^{n-2} \frac{1}{4} \frac{1}{\sqrt{\frac{p_{i+1}}{p_i}} + \sqrt{\frac{p_i}{p_{i+1}}}} \left[ -\cos\alpha + \cos((2i - 1)\alpha) \right].
\end{aligned}$$

According to (9), $e^{-1} \leq |p(j)/p(i)| \leq e$ for $|j - i| = 1$. We also have the inequality $\sqrt{\frac{p_j}{p_i}} + \sqrt{\frac{p_i}{p_j}} \geq 2$. Applying these bounds and separately summing terms of cosines with positive and negative signs, we

have an upper bound

$$\lambda_{\min} \leq v^\mathsf{T} A v \leq \frac{n-2}{n-1}\left(1 - \frac{1}{1+e}\right) - \frac{1}{n-1}(1 - \frac{1}{1+e}) \sum_{i=\lceil \frac{n-1}{4} \rceil}^{\lfloor \frac{3}{4}(n-1) \rfloor} \cos(2i\alpha)$$

$$- \frac{1}{n-1}\left(1 - \frac{1}{1+e^{-1}}\right)\left(\sum_{i=1}^{\lfloor \frac{n-1}{4} \rfloor} \cos(2i\alpha) + \sum_{i=\lceil \frac{3(n-1)}{4} \rceil}^{n-2} \cos(2i\alpha)\right)$$

$$- \frac{\cos\alpha}{2(n-1)}\frac{(n-2)}{\sqrt{e}}$$

$$+ \frac{1}{n-1}\frac{1}{2\sqrt{e}} \sum_{i=\lceil \frac{n+1}{4} \rceil}^{\lfloor \frac{3n-1}{4} \rfloor} \cos((2i-1)\alpha)$$

$$+ \frac{1}{n-1}\frac{1}{2}\left(\sum_{i=2}^{\lfloor \frac{n+1}{4} \rfloor} \cos((2i-1)\alpha) + \sum_{i=\lceil \frac{3n-1}{4} \rceil}^{n-2} \cos((2i-1)\alpha)\right)$$

$$+ \frac{1}{4(n-1)} \sum_{i=1,n} \cos((2i-1)\alpha).$$

Using exact formulas of sum of cosines, we have

$$\lambda_{\min} \leq v^\mathsf{T} A v \leq \frac{n-2}{n-1}\left(1 - \frac{1}{1+e}\right) - \frac{1}{n-1}\left(1 - \frac{1}{1+e}\right)\frac{\sin(\alpha a_{1n})}{\sin\alpha}\cos(\alpha b_{1n})$$

$$- \frac{1}{n-1}(1 - \frac{1}{1+e^{-1}})\left(\frac{\sin(\alpha a_{2n})}{\sin\alpha}\cos(\alpha b_{2n}) + \frac{\sin(\alpha a_{3n})}{\sin\alpha}\cos(\alpha b_{3n})\right)$$

$$- \frac{\cos\alpha}{2(n-1)}\frac{n-2}{\sqrt{e}}$$

$$+ \frac{1}{n-1}\frac{1}{2\sqrt{e}}\frac{\sin(\alpha a_{4n})}{\sin\alpha}\cos(\alpha b_{4n})$$

$$+ \frac{1}{n-1}\frac{1}{2}\left[\frac{\sin(\alpha a_{5n})}{\sin\alpha}\cos(\alpha b_{5n}) + \frac{\sin(\alpha a_{6n})}{\sin\alpha}\cos(\alpha b_{6n})\right]$$

$$+ \frac{1}{4(n-1)}\sum_{i=1,n}\cos((2i-1)\alpha),$$

with $a_{1n} = \lfloor \frac{3(n-1)}{4} \rfloor - \lceil \frac{n-1}{4} \rceil + 1$, $b_{1n} = \lfloor \frac{3(n-1)}{4} \rfloor + \lceil \frac{n-1}{4} \rceil$, $a_{2n} = \lfloor \frac{n-1}{4} \rfloor$, $b_{2n} = \lfloor \frac{n-1}{4} \rfloor + 1$, $a_{3n} = n - 1 - \lceil \frac{3(n-1)}{4} \rceil$, $b_{3n} = n - 2 + \lceil \frac{3(n-1)}{4} \rceil$, $a_{4n} = \lfloor \frac{3n-1}{4} \rfloor - \lceil \frac{n+1}{4} \rceil + 1$, $b_{4n} = \lfloor \frac{3n-1}{4} \rfloor + \lceil \frac{n+1}{4} \rceil - 1$, $a_{5n} = \lfloor \frac{n+1}{4} \rfloor - 1$, $b_{5n} = \lfloor \frac{n+1}{4} \rfloor + 1$, $a_{6n} = n - 1 - \lceil \frac{3n-1}{4} \rceil$, $b_{6n} = n - 3 + \lceil \frac{3n-1}{4} \rceil$. Note that $\alpha a_{1n}, \alpha a_{4n} \to \frac{\pi}{2}$ and $\alpha a_{in} \to \frac{\pi}{4}$ as $n \to \infty$ for all other $i$'s. Similarly, $\alpha b_{in}$ have corresponding limits. Also, $(n-1)\sin\alpha \to \pi$. Based on these limits, as $n \to \infty$, the right hand side of the inequality above also approaches a limit. Specifically in the limit,

$$\limsup_{n\to\infty} \lambda_{\min} \leq (1 - \frac{1}{1+e}) - \frac{1}{2\sqrt{e}} + (\frac{1}{2} - \frac{1}{1+e} + \frac{1}{1+e^{-1}} - \frac{1}{2\sqrt{e}})\frac{1}{\pi} \leq 0.64.$$

Therefore, $\liminf_{n\to\infty}\|I - S_D\|_2 \geq 1 - \limsup_{n\to\infty}\lambda_{\min} \geq 0.36$. This shows that for large $n$, $\|I - S_D\|_2$ is bounded away from 0. □

*Proof of* (10). We can find an explicit formula for $u$ by solving the linear system

$$\frac{p(i-1)}{2(p(i) + p(i-1))}(u(i) - u(i-1)) = \frac{p(i+1)}{2(p(i) + p(i+1))}(u(i+1) - u(i)) + 1, \quad i = 2, 3, \ldots, n-1$$

(14)

with $u(1) = u(n) = 0$. This yields

$$u(k) = 2 \left( \sum_{i=1}^{n-1} \frac{p(i) + p(i+1)}{p(i)p(i+1)} \right)^{-1} \left[ \left( \sum_{i=1}^{k-1} \frac{p(i) + p(i+1)}{p(i)p(i+1)} \right) \left( \sum_{i=k}^{n-1} \frac{p(i) + p(i+1)}{p(i)p(i+1)} \left( \sum_{\ell=2}^{i} p(\ell) \right) \right) \right.$$
$$\left. - \left( \sum_{i=k}^{n-1} \frac{p(i) + p(i+1)}{p(i)p(i+1)} \right) \left( \sum_{i=2}^{k-1} \frac{p(i) + p(i+1)}{p(i)p(i+1)} \left( \sum_{\ell=2}^{i} p(\ell) \right) \right) \right]. \tag{15}$$

By the symmetry of $p$,

$$u\left( \frac{n+1}{2} \right) = 2 \left( \sum_{i=1}^{n-1} \frac{p(i) + p(i+1)}{p(i)p(i+1)} \right)^{-1}$$
$$\left[ \left( \sum_{i=1}^{\frac{n+1}{2}-1} \frac{p(i) + p(i+1)}{p(i)p(i+1)} \right) \left( \sum_{i=\frac{n+1}{2}}^{n-1} \frac{p(i) + p(i+1)}{p(i)p(i+1)} \left( \sum_{\ell=2}^{i} p(\ell) \right) \right) \right.$$
$$\left. - \left( \sum_{i=\frac{n+1}{2}}^{n-1} \frac{p(i) + p(i+1)}{p(i)p(i+1)} \right) \left( \sum_{i=2}^{\frac{n+1}{2}-1} \frac{p(i) + p(i+1)}{p(i)p(i+1)} \left( \sum_{\ell=2}^{i} p(\ell) \right) \right) \right]$$

$$= 2 \left( \sum_{i=1}^{n-1} \frac{p(i) + p(i+1)}{p(i)p(i+1)} \right)^{-1}$$
$$\left[ \left( \sum_{i=1}^{\frac{n+1}{2}-1} \frac{p(i) + p(i+1)}{p(i)p(i+1)} \right) \left( \sum_{i=1}^{\frac{n-1}{2}} \frac{p(i) + p(i+1)}{p(i)p(i+1)} \left( \sum_{\ell=2}^{\frac{n+1}{2}} p(\ell) + \sum_{\ell=2}^{i} p(\ell) \right) \right) \right.$$
$$\left. - \left( \sum_{i=1}^{\frac{n+1}{2}-1} \frac{p(i) + p(i+1)}{p(i)p(i+1)} \right) \left( \sum_{i=2}^{\frac{n+1}{2}-1} \frac{p(i) + p(i+1)}{p(i)p(i+1)} \left( \sum_{\ell=2}^{i} p(\ell) \right) \right) \right]$$

$$= 2 \left( \sum_{i=1}^{n-1} \frac{p(i) + p(i+1)}{p(i)p(i+1)} \right)^{-1} \left[ \left( \sum_{i=1}^{\frac{n-1}{2}} \frac{p(i) + p(i+1)}{p(i)p(i+1)} \right)^2 \left( \sum_{\ell=2}^{\frac{n+1}{2}} p(\ell) \right) \right]$$

$$\geq \frac{1}{2} \left( \sum_{i=1}^{n-1} \frac{p(i) + p(i+1)}{p(i)p(i+1)} \right)^{-1} \left( \sum_{i=1}^{\frac{n-1}{2}} \frac{p(i) + p(i+1)}{p(i)p(i+1)} \right)^2$$

$$= \frac{1}{4} \left( \sum_{i=1}^{\frac{n-1}{2}} \frac{p(i) + p(i+1)}{p(i)p(i+1)} \right). \tag{16}$$

We can derive bounds based the last expression. A loose bound uses the fact that the sum is larger than the reciprocal of the smallest probability, and this probability is exponentially small.

$$u\left( \frac{n+1}{2} \right) \geq \frac{1}{4} \max_i \frac{1}{p(i)} \geq \frac{1}{4} \frac{3 \exp(\frac{n-1}{4\pi})}{\exp(-\frac{n-1}{8\pi})} = \frac{3}{4} \exp\left( \frac{3(n-1)}{8\pi} \right). \tag{17}$$

More carefully, for any small $\varepsilon > 0$, there are $\mathcal{O}(n)$, $\cos(\frac{4\pi(i-1)}{n-1})$ terms that are smaller than $-1 + \varepsilon$, and for these terms, $p(i) \leq \frac{1}{3} \exp(\frac{n-1}{4\pi}(-2 + \varepsilon))$. Therefore,

$$u\left( \frac{n+1}{2} \right) \gtrsim n \exp\left( \frac{n-1}{4\pi}(2 - \varepsilon) \right). \tag{18}$$

$\square$

*Proof of* (11). This time $u(i)$ solves

$$\frac{p(i-1)}{2(p(i)+p(i-1))}(u(i)-u(i-1)) = \frac{p(i+1)}{2(p(i)+p(i+1))}(u(i+1)-u(i)), \quad i=2,3,\ldots,n-1 \tag{19}$$

with $u(1) = 0$, $u(n) = 1$. We find that

$$u(2) = \frac{\frac{1}{p(1)} + \frac{1}{p(2)}}{\sum_{i=1}^{n-1} \frac{1}{p(i)} + \frac{1}{p(i+1)}}. \tag{20}$$

A loose bound is

$$u(2) \leq \frac{\frac{1}{p(1)} + \frac{1}{p(2)}}{\max_i \frac{1}{p(i)}} \leq \frac{\exp\left(-\frac{n-1}{4\pi}\right) + \exp\left(-\frac{n-1}{8\pi}\right)}{\exp\left(\frac{n-1}{8\pi}\right)} \lesssim \exp\left(-\frac{n-1}{4\pi}\right). \tag{21}$$

More carefully, for any small $\varepsilon > 0$, there are $\mathcal{O}(n)$, $\cos\left(\frac{4\pi(i-1)}{n-1}\right)$ terms that are smaller than $-1 + \varepsilon$, resulting in the bound

$$u(2) \lesssim \frac{\exp\left(-\frac{n-1}{8\pi}\right)}{n\exp\left(\frac{n-1}{4\pi}(1-\varepsilon)\right)} = \frac{1}{n}\exp\left(-\frac{(n-1)}{8\pi}(3-\varepsilon)\right). \tag{22}$$

$\square$

## C   Proofs of main results

*Proof of* (5) *in Theorem 1* . Each entry of $\tilde{S}^t - S^t$ is a ratio of two random variables, as

$$\tilde{S}^t(k,\ell) - S^t(k,\ell) = \frac{\frac{1}{M}\sum_{j=1}^M \mathbf{1}_{\{k\}}(X_0^j)\left(\mathbf{1}_{\{\ell\}}(X_{t\wedge T^j}^j) - S^t(k,\ell)\right)}{\frac{1}{M}\sum_{j=1}^M \mathbf{1}_{\{k\}}(X_0^j)} \tag{23}$$

We will prove all entries of $\tilde{S}^t - S^t$ for all $1 \leq t \leq \tau$ satisfy a central limit theorem by a central limit theorem for the (normalized) numerators, and then apply Slutsky's theorem to include the denominators. By an argument similar to the one used to establish Lemma 1 in Thiede et al. [2015], $u$ is a differentiable function of entries of $S^t$. The delta method, therefore, transforms a central limit theorem for these entries into a central limit theorem for $u$.

Let $\tilde{M}^t$ denote a matrix containing only the (normalized) numerators of $\tilde{S}^t - S^t$, that is, $\tilde{M}^t(k,\ell) = \frac{1}{M}\sum_{j=1}^M \mathbf{1}_{\{k\}}(X_0^j)\left(\mathbf{1}_{\{\ell\}}(X_{t\wedge T^j}^j) - S^t(k,\ell)\right)$. Let $\tilde{M}^{1:\tau}$ represent a long vector formed by flattening out each matrix $\tilde{M}^t$ with $1 \leq t \leq \tau$ in row major order and gluing up in time order. Then, since trajectories are independent, $\tilde{M}^{1:\tau}$ satisfies a central limit theorem as

$$\sqrt{M}\tilde{M}^{1:\tau} \xrightarrow{\mathcal{D}} \mathcal{N}(0, \Sigma_{\tilde{M}}^{1:\tau}). \tag{24}$$

For convenience, let $\Sigma_{\tilde{M}}^{1:\tau}((t_1,k_1,\ell_1),(t_2,k_2,\ell_2))$ denote the entry of $\Sigma_{\tilde{M}}^{1:\tau}$ that records the covariance

$$\mathbf{Cov}\left[\mathbf{1}_{\{k_1\}}(X_0)\left(\mathbf{1}_{\{\ell_1\}}(X_{t_1\wedge T}) - S^{t_1}(k_1,\ell_1)\right), \mathbf{1}_{\{k_2\}}(X_0)\left(\mathbf{1}_{\{\ell_2\}}(X_{t_2\wedge T}) - S^{t_2}(k_2,\ell_2)\right)\right]. \tag{25}$$

Since $\mathbf{1}_{\{k\}}(X_0)$ for different $k$ can not be nonzero at the same time, we immediately obtain

$$\Sigma_{\tilde{M}}^{1:\tau}((t_1,k_1,\ell_1),(t_2,k_2,\ell_2)) = 0. \quad \text{for } k_1 \neq k_2 \text{ and any } t_1,t_2,\ell_1,\ell_2. \tag{26}$$

Now we deal with the denominators. Let $\tilde{\mu}$ be the vector containing all the (normalized) denominators, that is, $\tilde{\mu}(k) = \frac{1}{M}M_k$. Then, again because of independence of trajectories, we have a weak law of large numbers as

$$\tilde{\mu} \xrightarrow{\mathcal{P}} \mu, \tag{27}$$

where we recall that $\mu$ is the initial distribution supported on $D$, and $\xrightarrow{\mathcal{P}}$ means convergence in probability.

Combining (23), (24), (26), and (27) , by Slutsky's theorem, we obtain a central limit theorem for $\tilde{S}$ as

$$\sqrt{M}\left(\tilde{S}^{1:\tau} - S^{1:\tau}\right) \xrightarrow{\mathcal{D}} \mathcal{N}(0, \Sigma^{1:\tau}), \tag{28}$$

where $\tilde{S}^{1:\tau}$ and $S^{1:\tau}$ are vectors flattened out from $\tilde{S}^t$ and $S^t$ for $1 \le t \le \tau$ with the same order as in $\tilde{M}^{1:\tau}$, and

$$\Sigma^{1:\tau}((t_1, k_1, \ell_1), (t_2, k_2, \ell_2)) = \frac{1}{\mu(k_1)\mu(k_2)} \Sigma_{\tilde{M}}^{1:\tau}((t_1, k_1, \ell_1), (t_2, k_2, \ell_2)). \tag{29}$$

According to (25) and (26),

$$\begin{aligned}
&\Sigma^{1:\tau}((t_1, k_1, \ell_1), (t_2, k_2, \ell_2)) = 0 \quad \text{for } k_1 \neq k_2, \\
&\Sigma^{1:\tau}((t_1, k, \ell_1), (t_2, k, \ell_2)) \\
&= \frac{1}{\mu(k)} \mathbf{E}_k \left[ \left( \mathbf{1}_{\{\ell_1\}}(X_{t_1 \wedge T}) - S^{t_1}(k_1, \ell_1) \right) \left( \mathbf{1}_{\{\ell_2\}}(X_{t_2 \wedge T}) - S^{t_2}(k_2, \ell_2) \right) \right].
\end{aligned} \tag{30}$$

To transform the central limit theorem (28) to a central limit theorem of $\tilde{u}(i)$, we point out that $u(i)$ is a differentiable function of entries of $S^t$ and now we are going to calculate these derivatives. Entries in $u$ outside of $D$ are fixed under any perturbation in $S^t$. To write down an equation satisfied by all entries of $u$, we rearrange (2) into

$$(I - S_D^\tau)u = \sum_{t=0}^{\tau-1} S^t R_D + (S^\tau - S_D^\tau) R_{D^c}, \tag{31}$$

where $R_{D^c}(i) = \mathbf{1}_{i \in D^c} R(i)$ for $i \in [n]$. $R_{D^c}$ is a vector whose nonzero entries are those fixed boundary values of $u$. Because $S^t$ is a stochastic matrix, we consider it to be determined by off diagonal entries as

$$S^t = I + \sum_{k \neq \ell, k \in D} S^t(k, \ell)(e_k e_\ell^\mathsf{T} - e_k e_k^\mathsf{T}). \tag{32}$$

This implies that

$$S_D^t = I + \sum_{k \in D, \ell \in D, k \neq \ell} S^t(k, \ell)(e_k e_\ell^\mathsf{T} - e_k e_k^\mathsf{T}) - \sum_{k \in D, \ell \in D^c} S^t(k, \ell) e_k e_k^\mathsf{T} - \sum_{k \in D^c} e_k e_k^\mathsf{T}. \tag{33}$$

For a specific pair $(k, \ell)$ with $k \in D$ and $\ell \neq k$, we define $S^t(\varepsilon) := S^t + \varepsilon(e_k e_\ell^\mathsf{T} - e_k e_k^\mathsf{T})$. Then we define

$$\frac{\partial u}{\partial S^t(k, \ell)}(S^t) := \left[ \frac{\partial u_1}{\partial S^t(k, \ell)}(S^t), \dots, \frac{\partial u_n}{\partial S^t(k, \ell)}(S^t) \right]^\mathsf{T} = \frac{d}{d\varepsilon} u\left(S^t(\varepsilon)\right)\Big|_{\varepsilon=0}.$$

Differentiating (31) with respect to $\varepsilon$ and taking care of (33), we find the derivative with respect to $S^t(k, \ell)$ for different $k, \ell, t$. The formulas are different for different $t$. For $t = \tau, k, \ell \in D$, we have

$$(I - S_D^\tau)\frac{\partial u}{\partial S^\tau(k, \ell)} - (e_k e_\ell^\mathsf{T} - e_k e_k^\mathsf{T})u = 0.$$

Therefore,

$$\frac{\partial u}{\partial S^\tau(k, \ell)} = (I - S_D^\tau)^{-1} e_k \left( u(\ell) - u(k) \right). \tag{34}$$

Instead, for $k \in D, \ell \in D^c$, taking derivatives leads to

$$(I - S_D^\tau)\frac{\partial u}{\partial S^\tau(k, \ell)} - e_k e_k^\mathsf{T} u = e_k e_\ell^\mathsf{T} R_{D^c},$$

which gives the same formula for $\frac{\partial u}{\partial S^\tau(k, \ell)}$ as in (34) after rearranging using the fact that $R_{D^c}(\ell) = u(\ell)$ for $\ell \in D^c$.

For $t < \tau$, $k \in D$ and $\ell \in [n]$, taking derivatives in (31) gives

$$\frac{\partial u}{\partial S^t(k,\ell)} = (I - S_D^\tau)^{-1} e_k \left(R_D(\ell) - R_D(k)\right). \tag{35}$$

Now, we can use the delta method to transform (28) to a central limit theorem for $\tilde{u}(i)$ for any $i \in D$.

$$\sqrt{M}\left(\tilde{u}(i) - u(i)\right) \xrightarrow{\mathcal{D}} \mathcal{N}(0, \sigma_i^2), \tag{36}$$

with

$$
\sigma_i^2 = \sum_{\substack{k_1,k_2 \in D, \\ \ell_1,\ell_2 \in [n], \\ k_1 \neq \ell_1, k_2 \neq \ell_2}} \sum_{t_1,t_2=1}^{\tau} \frac{\partial u(i)}{\partial S^{t_1}(k_1,\ell_1)} \frac{\partial u(i)}{\partial S^{t_2}(k_2,\ell_2)} \Sigma^{1:\tau}((t_1,k_1,\ell_1),(t_2,k_2,\ell_2))
$$

$$
= \sum_{k \in D} \frac{1}{\mu(k)} \left(e_i^\mathsf{T}(I - S_D^\tau)^{-1} e_k\right)^2 \mathbf{E}_k \left[\left(\left(\sum_{\substack{\ell \in [n] \\ \ell \neq k}} (R_D(\ell) - R_D(k)) \sum_{t=1}^{\tau-1}(\mathbf{1}_\ell(X_{t \wedge T}) - S^t(k,\ell))\right.\right.\right.
$$

$$
\left.\left.\left. + (u(\ell) - u(k))(\mathbf{1}_\ell(X_{\tau \wedge T}) - S^\tau(k,\ell))\right)^2\right]\right.
$$

$$
= \sum_{k \in D} \frac{1}{\mu(k)} \left(e_i^\mathsf{T}(I - S_D^\tau)^{-1} e_k\right)^2 \mathbf{E}_k \left[\left(\left(\sum_{\ell \in [n]} R_D(\ell) \sum_{t=1}^{\tau-1}(\mathbf{1}_\ell(X_{t \wedge T}) - S^t(k,\ell))\right.\right.\right.
$$

$$
\left.\left.\left. + u(\ell)(\mathbf{1}_\ell(X_{\tau \wedge T}) - S^\tau(k,\ell))\right)^2\right]\right.
$$

$$
= \sum_{k \in D} \frac{1}{\mu(k)} \left(e_i^\mathsf{T}(I - S_D^\tau)^{-1} e_k\right)^2 \mathbf{E}_k \left[\left(\sum_{t=1}^{\tau-1} R_D(X_{t \wedge T}) + u(X_{\tau \wedge T}) - (u(k) - R_D(k))\right)^2\right]
$$

$$
= \sum_{k \in D} \frac{1}{\mu(k)} \left(e_i^\mathsf{T}(I - S_D^\tau)^{-1} e_k\right)^2 \mathbf{E}_k \left[\left(\sum_{t=0}^{\tau-1} R_D(X_{t \wedge T}) + u(X_{\tau \wedge T}) - u(k)\right)^2\right]. \tag{37}
$$

The second equality is obtained by plugging in (35), (34) and (30). The third equality uses the fact that

$$\sum_{\ell \in [n]} \mathbf{1}_\ell(X_{t \wedge T}) = \sum_{\ell \in [n]} S^t(k,\ell) = 1, \text{ for any } k \in D, t \leq \tau.$$

The fourth equality uses the Bellman equation (2). □

Now we introduce the key lemma for Theorem 1. The lemma characterizes the continuity of the operator $\mathbf{E}_\ell\left[\sum_{t=0}^T \mathbf{1}_{\{j\}}(X_t)\right]$ in terms of the initial state $\ell$. Note that a dot product of this operator with the corresponding reward $R(j)$ generates the value function $u(\ell)$.

We define additional hitting times counting from $t = 0$. That is, $\hat{T}_k = \min\{t \geq 0 : X_t = k\}$. This differs from $T_k$ because $\hat{T}_k$ includes $t = 0$. Similarly, we define $\hat{T}_k^\tau$ and more generally $\hat{T}_A$ and $\hat{T}_A^\tau$ for any subset $A \subseteq [n]$ as the counterparts of $T_k^\tau$, $T_A$, and $T_A^\tau$ but counted from $t = 0$. But notice that we have defined $T$ and $T^\tau$ to be counted from $t = 0$. In all hitting time definitions, when the set over which the minimum is taken is empty, the hitting time is defined to be $+\infty$.

**Lemma 2.** *For $i, k \in D$, $\ell, j \in [n]$, $k \neq \ell$,*

$$\mathbf{E}_i\left[\sum_{t=0}^{T^\tau-1} \mathbf{1}_{\{k\}}(Y_t^\tau)\right]\left|\mathbf{E}_\ell\left[\sum_{t=0}^T \mathbf{1}_{\{j\}}(X_t)\right] - \mathbf{E}_k\left[\sum_{t=0}^T \mathbf{1}_{\{j\}}(X_t)\right]\right| \leq \frac{\mathbf{E}_i\left[\sum_{t=0}^T \mathbf{1}_{\{j\}}(X_t)\right]}{Q^\tau(k,\ell)}. \tag{38}$$

*Proof of Lemma 2.* The proof will proceed by representing all quantities by probabilities, and then with the help of a complementary Markov chain, we can bound the difference of involved probabilities. We start with two basic identities that hold for any $i, k \in D$,

$$\mathbf{E}_i \left[ \sum_{t=0}^{T^\tau - 1} \mathbf{1}_{\{k\}} \left( Y_t^\tau \right) \right] = \frac{\mathbf{P}_i \left[ \hat{T}_k^\tau < T^\tau \right]}{\mathbf{P}_k \left[ T^\tau < T_k^\tau \right]}, \tag{39}$$

and

$$\mathbf{E}_\ell \left[ \sum_{t=0}^{T} \mathbf{1}_{\{j\}} \left( X_t \right) \right] = \frac{\mathbf{P}_\ell \left[ \hat{T}_j \le T \right]}{\mathbf{P}_j \left[ T \le T_j \right]}. \tag{40}$$

Note that when $j \in D^c$, $\mathbf{P}_j \left[ T \le T_j \right] = 1$. With (39) and (40), the left hand side of (38) becomes

$$\frac{\mathbf{P}_i \left[ \hat{T}_k^\tau < T^\tau \right]}{\mathbf{P}_k \left[ T^\tau < T_k^\tau \right]} \left| \frac{\mathbf{P}_\ell \left[ \hat{T}_j \le T \right] - \mathbf{P}_k \left[ \hat{T}_j \le T \right]}{\mathbf{P}_j \left[ T \le T_j \right]} \right|. \tag{41}$$

To make probabilities related to $X_t$ in (41) become probabilities related to $\tau$-step transitions, we consider another Markov chain $Y_t^{\tau,j}$ that follows the transition probability matrix $S_j^\tau := (S_j)^\tau$, where $S_j$ is defined as

$$S_j(k, \ell) = \begin{cases} S(k, \ell) & k \ne j, k, \ell \in [n] \\ \delta_{k\ell} & k = j, \ell \in [n]. \end{cases} \tag{42}$$

$S_j$ is the same as $S$ except it is stopped once $j$ is hit. Note that when $j \in D^c$, $S_j = S$. Define hitting times $T_k^{\tau,j}$, $T^{\tau,j}$ and $\hat{T}_k^{\tau,j}$ of $Y_t^{\tau,j}$ similarly as those of $Y_t^\tau$. If $Y_t^{\tau,j}$ hits $j$ before $k$, then $T_k^{\tau,j} = +\infty$, and this exception is similarly defined for other hitting times. Note that $T^{\tau,j}$ counts from $t = 0$ as does $T$. One key relation between $Y_t^{\tau,j}$ and $X_{t \wedge T}$ is that

$$\mathbf{P}_\ell \left[ \hat{T}_j \le T \right] = \mathbf{P}_\ell \left[ \hat{T}_j^{\tau,j} \le T^{\tau,j} \right]. \tag{43}$$

This relation is guaranteed by the fact that $Y_t^{\tau,j}$ will stay at $j$ if a coupled $X_{t \wedge T}$ hits $j$ at any time within each length-$\tau$ time interval.

Now consider the case where $\ell \in D$, $j \ne k$ and $j \ne \ell$, and identify all of $D^c \setminus \{j\}$ with a single additional index $\Delta$. Note that whether $j \in D$ or $j \in D^c$,

$$\mathbf{P}_\ell \left[ \hat{T}_j^{\tau,j} \le T^{\tau,j} \right] = \mathbf{P}_\ell \left[ \hat{T}_j^{\tau,j} < T_\Delta^{\tau,j} \right]. \tag{44}$$

Consider another Markov chain $Z_t \in \{\ell, k, j, \Delta\}$ that records transitions only when $Y_t^{\tau,j}$ transitions between one of these states. We will use $Q$ to denote the transition probabilities of $Z_t$. For example,

$$Q(\ell, \ell) = \mathbf{P}_\ell \left[ Z_1 = \ell \right] = \mathbf{P}_\ell \left[ T_\ell^{\tau,j} < \min\{T_k^{\tau,j}, T_j^{\tau,j}, T_\Delta^{\tau,j}\} \right]$$

and

$$Q(\ell, k) = \mathbf{P}_\ell \left[ Z_1 = k \right] = \mathbf{P}_\ell \left[ T_k^{\tau,j} < \min\{T_\ell^{\tau,j}, T_j^{\tau,j}, T_\Delta^{\tau,j}\} \right].$$

In fact, $\mathbf{P}_\ell \left[ T_j^{\tau,j} < T_\Delta^{\tau,j} \right]$ and $\mathbf{P}_k \left[ T_j^{\tau,j} < T_\Delta^{\tau,j} \right]$ satisfy the linear system

$$\mathbf{P}_\ell \left[ T_j^{\tau,j} < T_\Delta^{\tau,j} \right] = Q(\ell, \ell) \mathbf{P}_\ell \left[ T_j^{\tau,j} < T_\Delta^{\tau,j} \right] + Q(\ell, k) \mathbf{P}_k \left[ T_j^{\tau,j} < T_\Delta^{\tau,j} \right] + Q(\ell, j),$$
$$\mathbf{P}_k \left[ T_j^{\tau,j} < T_\Delta^{\tau,j} \right] = Q(k, k) \mathbf{P}_k \left[ T_j^{\tau,j} < T_\Delta^{\tau,j} \right] + Q(k, \ell) \mathbf{P}_\ell \left[ T_j^{\tau,j} < T_\Delta^{\tau,j} \right] + Q(k, j). \tag{45}$$

It is convenient to work with the normalized transition probabilities

$$\overline{Q}(\ell, \cdot) = \frac{Q(\ell, \cdot)}{1 - Q(\ell, \ell)} \quad \text{and} \quad \overline{Q}(k, \cdot) = \frac{Q(k, \cdot)}{1 - Q(k, k)}.$$

With this notation, solving (45) reveals the identities

$$\mathbf{P}_\ell \left[ T_j^{\tau,j} < T_\Delta^{\tau,j} \right] = \frac{\overline{Q}(\ell, k) \overline{Q}(k, j) + \overline{Q}(\ell, j)}{1 - \overline{Q}(\ell, k) \overline{Q}(k, \ell)}$$

and

$$\mathbf{P}_k\left[T_j^{\tau,j} < T_\Delta^{\tau,j}\right] = \frac{\overline{Q}(k,\ell)\overline{Q}(\ell,j) + \overline{Q}(k,j)}{1 - \overline{Q}(\ell,k)\overline{Q}(k,\ell)},$$

from which, after a little more algebra, we find that

$$\mathbf{P}_\ell\left[T_j^{\tau,j} < T_\Delta^{\tau,j}\right] - \mathbf{P}_k\left[T_j^{\tau,j} < T_\Delta^{\tau,j}\right] = \frac{-\overline{Q}(\ell,\Delta)\overline{Q}(k,j) + \overline{Q}(k,\Delta)\overline{Q}(\ell,j)}{1 - \overline{Q}(\ell,k)\overline{Q}(k,\ell)}. \qquad (46)$$

This is the key quantity that characterizes the difference between probabilities with initial states $k$ and $\ell$. By bounding this, we can finally get relative variance bounds. We observe immediately that

$$\frac{\overline{Q}(\ell,\Delta)\overline{Q}(k,j)}{1 - \overline{Q}(\ell,k)\overline{Q}(k,\ell)} \le \mathbf{P}_k\left[T_j^{\tau,j} < T_\Delta^{\tau,j}\right]\overline{Q}(\ell,\Delta).$$

Now we relate these probabilities to $Y_t^\tau$. We have that $\mathbf{P}_k\left[T_\ell^\tau < T_k^\tau \wedge T^\tau\right] \le Q(k,\ell) + Q(k,j)$, so

$$\frac{Q(k,\ell)\overline{Q}(\ell,j) + Q(k,j)}{\mathbf{P}_k\left[T_\ell^\tau < T_k^\tau \wedge T^\tau\right]} \ge \frac{\overline{Q}(k,\ell)\overline{Q}(\ell,j) + \overline{Q}(k,j)}{\overline{Q}(k,\ell) + \overline{Q}(k,j)} \ge \overline{Q}(\ell,j).$$

This yields

$$\frac{\overline{Q}(k,\Delta)\overline{Q}(\ell,j)}{1 - \overline{Q}(\ell,k)\overline{Q}(k,\ell)} \le \frac{\mathbf{P}_k\left[T_j^{\tau,j} < T_\Delta^{\tau,j}\right]Q(k,\Delta)}{\mathbf{P}_k\left[T_\ell^\tau < T_k^\tau \wedge T^\tau\right]}.$$

Bringing these inequalities back into (46), with (43) and (44) we have

$$\left|\mathbf{P}_\ell\left[\hat{T}_j \le T\right] - \mathbf{P}_k\left[\hat{T}_j \le T\right]\right| \le \mathbf{P}_k\left[T_j^{\tau,j} < T_\Delta^{\tau,j}\right]\max\left\{\overline{Q}(\ell,\Delta), \frac{Q(k,\Delta)}{\mathbf{P}_k\left[T_\ell^\tau < T_k^\tau \wedge T^\tau\right]}\right\}.$$

Now notice that $\mathbf{P}_k\left[T^\tau < T_k^\tau\right] \ge Q(k,\Delta) + \mathbf{P}_k\left[T_\ell^\tau < T_k^\tau \wedge T^\tau\right]\overline{Q}(\ell,\Delta)$, and therefore,

$$\frac{\left|\mathbf{P}_\ell\left[\hat{T}_j^{\tau,j} \le T^{\tau,j}\right] - \mathbf{P}_k\left[\hat{T}_j^{\tau,j} \le T^{\tau,j}\right]\right|}{\mathbf{P}_k\left[T^\tau < T_k^\tau\right]} \le \frac{\mathbf{P}_k\left[T_j^{\tau,j} \le T^{\tau,j}\right]}{\mathbf{P}_k\left[T_\ell^\tau < T_k^\tau \wedge T^\tau\right]}. \qquad (47)$$

Plugging (47) back into (41) we find that

$$\mathbf{E}_i\left[\sum_{t=0}^{T^\tau - 1}\mathbf{1}_{\{k\}}(Y_t^\tau)\right]\left|\mathbf{E}_\ell\left[\sum_{t=0}^{T}\mathbf{1}_{\{j\}}(X_t)\right] - \mathbf{E}_k\left[\sum_{t=0}^{T}\mathbf{1}_{\{j\}}(X_t)\right]\right|$$

$$\le \frac{\mathbf{P}_i\left[\hat{T}_k^\tau < T^\tau\right]\mathbf{P}_k\left[T_j^{\tau,j} \le T^{\tau,j}\right]}{\mathbf{P}_j\left[T \le T_j\right]\mathbf{P}_k\left[T_\ell^\tau < T_k^\tau \wedge T^\tau\right]} \le \frac{\mathbf{P}_i\left[\hat{T}_j \le T\right]}{\mathbf{P}_j\left[T \le T_j\right]\mathbf{P}_k\left[T_\ell^\tau < T_k^\tau \wedge T^\tau\right]}. \qquad (48)$$

By (40) and the definition of $Q^\tau(k,\ell)$ in Theorem 1, this is the stated bound in the case where $\ell \in D, j \ne k$ and $j \ne \ell$.

Now we deal with the exceptional cases. When $\ell \in D$ but $j = k$ or $j = \ell$, the same conclusion holds with minor modifications of the intermediate steps. Specifically, take $j = \ell$ (so that $j \in D$ and $\Delta = D^c$) as an example. Then $Z_t \in \{\ell, k, \Delta\}$ records transitions of $Y_t^{\tau,j}$ among these three states. With $Q$ denoting its transition probabilities.

$$\mathbf{P}_\ell\left[\hat{T}_j^{\tau,j} \le T^{\tau,j}\right] - \mathbf{P}_k\left[\hat{T}_j^{\tau,j} \le T^{\tau,j}\right] = 1 - \overline{Q}(k,j) = 1 - \overline{Q}(k,\ell) = \overline{Q}(k,\Delta),$$

and $\mathbf{P}_k\left[T_\ell^\tau < T_k^\tau \wedge T^\tau\right] = Q(k,\ell), \mathbf{P}_k\left[T_j < T\right] = \overline{Q}(k,\ell)$. Therefore,

$$\left|\mathbf{P}_\ell\left[\hat{T}_j \le T\right] - \mathbf{P}_k\left[\hat{T}_j \le T\right]\right| \le \frac{\mathbf{P}_k\left[T_j < T\right]Q(k,\Delta)}{\mathbf{P}_k\left[T_\ell^\tau < T_k^\tau \wedge T^\tau\right]}.$$

We also note that $\mathbf{P}_k\left[T^\tau < T_k^\tau\right] \ge Q(k,\Delta)$. Then the following steps are the same as (47) and (48) in the general case.

When $\ell \in D^c$, $j \in [n]$ and $\ell \in \Delta$, $\mathbf{P}_\ell\left[\hat{T}_j \le T\right] = 0$.

$$\frac{\mathbf{P}_i\left[\hat{T}_k^\tau < T^\tau\right]}{\mathbf{P}_k\left[T^\tau < T_k^\tau\right]}\frac{\left|\mathbf{P}_\ell\left[\hat{T}_j \le T\right] - \mathbf{P}_k\left[\hat{T}_j \le T\right]\right|}{\mathbf{P}_j\left[T \le T_j\right]} = \frac{\mathbf{P}_i\left[\hat{T}_k^\tau < T^\tau\right]\mathbf{P}_k\left[\hat{T}_j \le T\right]}{\mathbf{P}_k\left[T^\tau < T_k^\tau\right]\mathbf{P}_j\left[T \le T_j\right]}$$

$$\le \frac{\mathbf{P}_i\left[\hat{T}_j \le T\right]}{\mathbf{P}_j\left[T \le T_j\right]\mathbf{P}_k\left[T_\ell^\tau < T_k^\tau \wedge T_{D^c\setminus\{\ell\}}^\tau\right]}.$$

Applying (40) to the right hand side, this gives the bound (38).

When $\ell \in D^c$ and $j = \ell$, $\mathbf{P}_\ell\left[\hat{T}_j \le T\right] = 1$. We again use $Q$ to record transitions of $Y_t^{\tau,j}(= Y_t^\tau)$ among $\{k, j, \Delta\}$. Then,

$$\frac{\mathbf{P}_i\left[\hat{T}_k^\tau < T^\tau\right]}{\mathbf{P}_k\left[T^\tau < T_k^\tau\right]}\frac{\left|\mathbf{P}_\ell\left[\hat{T}_j \le T\right] - \mathbf{P}_k\left[\hat{T}_j \le T\right]\right|}{\mathbf{P}_j\left[T \le T_j\right]} \le \frac{\mathbf{P}_i\left[\hat{T}_k^\tau < T^\tau\right]}{\mathbf{P}_j\left[T \le T_j\right]\mathbf{P}_k\left[T^\tau < T_k^\tau\right]}$$

$$\le \frac{\mathbf{P}_i\left[\hat{T}_j \le T\right]}{\mathbf{P}_j\left[T \le T_j\right]\mathbf{P}_k\left[T_\ell^\tau < T_k^\tau \wedge T_{D^c\setminus\{\ell\}}^\tau\right]}.$$

The last inequality follows from the facts that

$$\frac{1}{\mathbf{P}_k\left[T^\tau < T_k^\tau\right]} = \frac{1}{1 - Q(k,k)} = \frac{\overline{Q}(k,j)}{Q(k,j)} = \frac{\mathbf{P}_k\left[T_j^\tau \le T^\tau\right]}{\mathbf{P}_k\left[T_\ell^\tau < T_k^\tau \wedge T_{D^c\setminus\{\ell\}}^\tau\right]}.$$

We thus prove that the bound (38) holds for any indices $i, k, j \in D$, $\ell \in [n]$ and $k \ne \ell$. $\qquad\square$

Now we are well prepared to prove (6) in Theorem 1.

*Proof of* (6) *in Theorem 1.* We will interpret two main terms in (5) probabilistically. First, for $i, k \in D$,

$$e_i^\mathsf{T}(I - S_D^\tau)^{-1}e_k = \sum_{t=0}^\infty (S_D^\tau)^t(i,k) = \mathbf{E}_i\left[\sum_{t=0}^{T^\tau-1}\mathbf{1}_{\{k\}}(Y_t^\tau)\right]. \qquad (49)$$

As for the expectation following $e_i^\mathsf{T}(I - S_D^\tau)^{-1}e_k$ in (5), we note from the Bellman equation (2) that

$$\mathbf{E}_k\left[\sum_{t=0}^{\tau-1}R_D(X_{t\wedge T}) + u(X_{\tau\wedge T}) - u(k)\right]^2 = \mathbf{Var}_k\left[\sum_{t=1}^{\tau-1}R_D(X_{t\wedge T}) + u(X_{\tau\wedge T})\right],$$

where the subscript $k$ indicates conditioning on $X_0 = k$. Since this is a variance, we can decompose it into conditional variance of each step by the law of total variance, as

$$\mathbf{Var}_k\left[\sum_{t=1}^{\tau-1}R_D(X_{t\wedge T}) + u(T_{\tau\wedge T})\right] = \mathbf{E}_k\left[\mathbf{Var}\left(\sum_{t=1}^{\tau-1}R_D(X_{t\wedge T}) + u(X_{\tau\wedge T})\,\Big|\,X_0 = k, X_{1\wedge T}\right)\right]$$

$$+ \mathbf{Var}_k\left[\mathbf{E}\left(\sum_{t=1}^{\tau-1}R_D(X_{t\wedge T}) + u(X_{\tau\wedge T})\,\Big|\,X_0 = k, X_{1\wedge T}\right)\right].$$

$$(50)$$

By the definition of $u$, $\mathbf{E}\left(\sum_{t=1}^{\tau-1}R_D(X_{t\wedge T}) + u(X_{\tau\wedge T})\,\Big|\,X_0 = k, X_{1\wedge T}\right) = u(X_{1\wedge T})$, which helps to simplify the second term. Meanwhile, the first term can be simplified as

$$\mathbf{E}_k\left[\mathbf{Var}\left(\sum_{t=1}^{\tau-1}R_D(X_{t\wedge T}) + u(X_{\tau\wedge T})\,\Big|\,X_0 = k, X_{1\wedge T}\right)\right]$$

$$= \mathbf{E}_k\left[\mathbf{Var}\left(\sum_{t=2}^{\tau-1}R_D(X_{t\wedge T}) + u(X_{\tau\wedge T})\,\Big|\,X_0 = k, X_{1\wedge T}\right)\right].$$

We can further condition it on $X_{2 \wedge T}$ and perform a similar decomposition as in (50), which can be done inductively for $X_{t \wedge T}$ up to $t = \tau$. As a result of this inductive conditioning, we obtain

$$
\begin{aligned}
\mathbf{E}_k &\left[ \sum_{t=0}^{\tau-1} R_D(X_{t \wedge T}) + u(X_{\tau \wedge T}) - u(k) \right]^2 \\
=& \mathbf{E}_k(\mathbf{Var}_{X_{(\tau-1)\wedge T}}(u(X_{\tau \wedge T}))) + \cdots + \mathbf{E}_k(\mathbf{Var}_{X_{2 \wedge T}}(u(X_{3 \wedge T}))) + \mathbf{E}_k(\mathbf{Var}_{X_{1 \wedge T}}(u(X_{2 \wedge T}))) \\
& + \mathbf{Var}_k(u(X_{1 \wedge T})) \\
\leq & \sum_{\substack{\ell \in [n] \\ \ell \neq k}} (u(\ell) - u(k))^2 \left[ \sum_{t=1}^{\tau} S^t(k, \ell) \right].
\end{aligned}
$$

(51)

For the inequality, we use the property of variance that

$$
\mathbf{E}_k(\mathbf{Var}_{X_{(t-1)\wedge T}}(u(X_{t \wedge T}))) \leq \mathbf{E}_k(\mathbf{E}_{X_{(t-1)\wedge T}}(u(X_{t \wedge T}) - u(k))^2) = \mathbf{E}_k(u(X_{t \wedge T}) - u(k))^2.
$$

Now, we further decompose the difference $u(\ell) - u(k)$ into components that we have already bounded in Lemma 2, using the fact that $R$ is nonnegative throughout our paper.

$$
\begin{aligned}
|u(\ell) - u(k)| &= \left| \sum_{j \in [n]} \left[ \mathbf{E}_\ell \left( \sum_{t=0}^{T} \mathbf{1}_{\{j\}}(X_t) R(j) \right) - \mathbf{E}_k \left( \sum_{t=0}^{T} \mathbf{1}_{\{j\}}(X_t) R(j) \right) \right] \right| \\
&\leq \sum_{j \in [n]} \left| \mathbf{E}_\ell \left( \sum_{t=0}^{T} \mathbf{1}_j(X_t) \right) - \mathbf{E}_k \left( \sum_{t=0}^{T} \mathbf{1}_j(X_t) \right) \right| R(j).
\end{aligned}
$$

(52)

Then according to Lemma 2, we can bound the variation in $u$ between states as

$$
\mathbf{E}_i \left[ \sum_{t=0}^{T^\tau - 1} \mathbf{1}_{\{k\}}(Y_t^\tau) \right] |u(\ell) - u(k)| \leq \sum_{j \in [n]} \frac{\mathbf{E}_i \left( \sum_{t=0}^{T} \mathbf{1}_{\{j\}}(X_t) \right) R(j)}{Q^\tau(k, \ell)} = \frac{u(i)}{Q^\tau(k, \ell)}.
$$

Combined with (49) and (51), this yields

$$
\sigma_i^2 \leq u^2(i) \sum_{k \in D} \frac{1}{\mu(k)} \frac{\sum_{t=1}^{\tau} S^t(k, \ell)}{Q^\tau(k, \ell)^2}. \qquad \square
$$

As for Corollary 1, the strengthening comes from an intermediate step in the proof of (6).

*Proof of Corollary 1.* If $R(i) = 0$ for any $i \in D$ (for example, when $u$ is the committor function), then

$$
\mathbf{E}_k \left( \sum_{t=0}^{\tau-1} R_D(X_{t \wedge T}) + u(X_{\tau \wedge T}) - u(k) \right)^2 = \sum_{\substack{\ell \in [n] \\ \ell \neq k}} (u(\ell) - u(k))^2 S^\tau(k, \ell). \qquad (53)
$$

Compared with (51), we see that the same following steps will replace $\sum_t S^t(k, \ell)$ in Theorem 1 by $S^\tau$ for Corollary 1. $\qquad \square$

