# OpenReview forum: "The surprising efficiency of temporal difference learning for rare event prediction"
_NeurIPS.cc/2024/Conference — NeurIPS 2024 poster_

### Official Review · Reviewer_piRP · 2024-07-04

**Soundness:** 4
**Presentation:** 4
**Contribution:** 4
**Rating:** 7
**Confidence:** 3

**Summary:**

The paper studies the estimation of rare event statistics in discrete-time, discrete-state Markov chains. It shows that when the transitions probabilities of the chain satisfy certain locality assumptions then temporal difference estimators are exponentially more efficient than Monte-Carlo estimators. Concretely, under these assumptions, the variance of the TD estimator scales as $n^3$ in the number $n$ of states of the Markov chain, whereas the variance for MC estimators scales as $e^n$.

The proofs are accompanied by two numerical examples that provide intuition and illustrate the applicability of the theory to mean first passage time and committor function computation. They also provide an additional validation of the theoretical results.

**Strengths:**

Overall great paper. The setting is sharp and precisely captures the phenomenon to be studied. The paper relies on a key original insight that links locality of transitions and efficiency of TD estimators for rare event estimation.

The related work is clearly covered, and the paper convincingly distills the key insights from previous empirical and theoretical studies into a set of assumptions and their consequences in the setting of an asymptotic central limit theorem.

The setting is consistent with standard notions of Markov chain literature. The key ideas are nicely illustrated using examples and pictures. Presentation is sharp overall.

The clear understanding of why and when TD performs well allow practitioners to make an informed choice on when to use TD for sampling problems.

**Weaknesses:**

The problem of rare event estimation is essentially a statistics problem, and the results seem more typical of a probability or statistics journal paper than a machine learning conference.

**Questions:**

No questions at this point.

**Limitations:**

In general, the theoretical nature of the paper and clear exposition make most of the limitations appear clearly in the statements of the theorems.

---

> ### Author Rebuttal · Authors · 2024-08-06
>
> Thank you very much for your positive feedback and acknowledgement of our work’s contributions. While our specific contributions are statistical in nature, we feel they are very relevant to machine learning based prediction for rare events. For example, in the context of extreme event prediction for scientific applications, machine learning procedures based on the MC estimator are common. Our results help to explain the relative effectiveness of recent TD based rare event prediction schemes cited in the manuscript. We hope to build a bridge between the scientific community, which studies rare events, and the RL community, which seeks theoretical foundations for the benefits of TD. This interdisciplinary focus also motivated our decision to submit to NeurIPS rather than a statistics journal.

---

### Official Review · Reviewer_251G · 2024-07-13

**Soundness:** 2
**Presentation:** 2
**Contribution:** 2
**Rating:** 5
**Confidence:** 2

**Summary:**

The authors consider the problem of efficiency comparison between temporal difference (TD) and Monte Carlo (MC) methods for learning value functions (policy evaluation) especially in conditions involving rare events. The authors identify two challenges in policy evaluation with respect to rare events - long timescales of the rare event and need for relative accuracy in estimates of very small values. For their analysis, the authors choose the LSTD algorithm and upper bound its relative asymptotic variance. Using this bound they establish the better efficiency of TD versus MC methods in the presence of rare events. Specifically teh authros define rare events in terms of escape from a given set of states in a Markov reward process.

**Strengths:**

The paper is mostly clear and the problem selected is an interesting and relevant one.

**Weaknesses:**

The authors compare LSTD and MC methods for policy estimation in the presence of rare events, but do not present any numerical experiments to illustrate their claims. It is not clear how the results of this paper can be practically used.

**Questions:**

1. What do the authors mean by "appropriate choice of reward function" in the statement in line 49: "Rare events can be embedded in such a formulation either as “escape from D,” or as “escape from D in an unusual way,” by appropriate choice of the reward function."
2. Is $T$ a terminal time value for the process as given in equation (1)? Is it assumed that the process ends as soon as the trajectory leaves the set $D$?
3. How does $u(i)$ reflect/determine a rare event? Can't two states resulting in two trajectories - one with a rare event and one without have the same value $u$?
4. Can the authors elaborate this point in line 68: "The MC estimator does not allow finite values of $\tau$ (unless $T$ is bounded by a constant).". Will this point change in discounted return settings?
4. $S$ is used in footnote 1, but defined later. Can the authors fix this?
5. How have the authors addressed the bias-variance tradeoff in TD vs MC methods?
6. As per the expressions given after line 87, are all states $i \notin D$ stationary states?
7. I did not understand the experiments given in Section 3. Don't escape events need to be observed in both TD and MC cases for the algorithm to evaluate the value function accurately? Don't TD and MC only vary in their ability to generalize as TD makes a Markovian assumption, while MC does not? Do the examples mainly illustrate this point?
8. What errors does the minorizing step in Section 4 introduce?

**Limitations:**

The authors mention the limitations in their work as points in different sections but do not provide a separate section only discussing these.

---

> ### Author Rebuttal · Authors · 2024-08-06
>
> We thank the reviewer for the opportunity to respond to detailed questions. We respond to specific comments below. Space for our responses is limited, so please let us know if you have additional questions.
>
> **Weaknesses**:
>
> For a detailed response regarding our numerical experiments, please see our General Response.
>
> Regarding practical applications of our results, we hope our paper will prompt a greater focus on rare events and their challenges within the reinforcement learning community as well as highlight the potential exponential benefits of TD over MC in those scenarios. As in our related work section, although TD has been a central idea in RL, quantitative characterizations of its benefits over MC have been lacking. Meanwhile, in extreme event prediction for scientific applications, machine learning procedures based on the MC estimator are common. By characterizing scenarios where TD excels, our paper can guide practitioners to make informed decisions about when to use TD and how to tune parameters such as the sample size and lag time according to the problem.
>
> **Questions**:
>
> 1. The statement at line 49 is intended to convey that we consider both cases in which the escape time from $D$ is very large and cases in which only low probability escape paths accumulate significant values of $R$. Both are cases in which the MC estimator is very inefficient. We will state this more directly. Concrete examples of $D$ and $R$ corresponding to common rare event statistics are given in Sec. 3. In the mean first passage time estimation problem in Sec. 3, we do not distinguish between different escape paths. In the committor estimation problem we estimate the probability of escape at $n$ (and not at 1). These two problems illustrate the essential difficulties.
> 2. Yes, throughout the paper $T$ is the stopping time defined after eq. (1). Please also see our answer to Question 7.
> 3. Please also refer to our answer to Question 1. It is possible to have very large (small) values of $u$ without any rare event by simply choosing $R$ to be large (small). It is the variation in the accumulated reward over different possible trajectories that leads to large error for the MC estimator.
> 4. In our setting the MC trajectories must extend all the way until time $T$ in order to construct the MC estimator (lines 59-60). With a discount factor of $\gamma$, the value function can again be written as the expected accumulated reward up to a geometrically distributed time, $T$, with mean ($1/(1-\gamma)$). An unbiased MC estimator would again require trajectories that extend to this time. In either setting, limiting the length of trajectories will introduce bias (even in the large $M$ limit) to the MC estimate.
> 5. Thanks! We will address this.
> 6. Since the bias of TD in our finite state setting decreases with $1/M$, for large $M$, the MSE of TD is dominated by its variance. The MC estimator is unbiased, so we characterize only its asymptotic variance. Since the bias of the TD estimator decreases as the lag time $\tau$ increases (i.e. as TD approaches MC), one could consider the bias variance tradeoff as a function of $\tau$ with finite $M$. This is an interesting question, but requires non-asymptotic bounds for bias and variance to address. The overlap of empirical relative MSE and asymptotic variance in the middle panels of Figs. 1 and 2, strongly suggests that the bias of TD is dominated by its variance in the rare event setting.
> 7. The transition matrix of $X_t$ is $P$ (not $S$) and $X_t$ need not stop when it escapes $D$. However, $u$ only includes rewards accumulated up to escape. So, for the purpose of constructing any estimator of $u$, we can regard states outside of $D$ as stationary. This leads to the Bellman equation eq. (2) upon which the TD estimator is based.
> 8. TD does require escape events in its data set. The data set used in Sec. 3 uses initial conditions drawn from the uniform distribution. Some trajectories started near states 1 or $n$ do escape before time $\tau$. However, most trajectories in the TD data set need not escape, and those that do are of length no longer than $\tau$. In MC every sample trajectory needs to escape and the trajectories can be of arbitrary length. Applying the MC estimator using only trajectories that escape within time $\tau$, would produce very biased estimates (the estimate of the mean first passage time would always be less than $\tau$). Understanding the performance advantages of TD over MC is a long standing challenge. While the TD estimator does exploit Markovianity, the key to TD’s dramatic performance advantage in the rare event setting (and in particular in the numerical experiments in Sec. 3) is that it uses estimates of many relatively large quantities (matrix entries referenced in Assumption 2) to estimate rare event statistics. Assumption 2 ensures that this is possible, as discussed more in our response to Question 9. We will provide this intuition more explicitly in Sec. 4.
> 9. The minorization referred to in Assumption 2 is not an approximation. It stipulates that any two states can be connected by a sequence of transitions, each of which has, individually, significant probability, even if transitions between the two states are very rare. As our examples in Sec. 3 demonstrate, this assumption does not at all rule out rare events. Returning to our answer to Question 9, Assumption 2 ensures that the TD estimator implicitly expresses the rare event statistic of interest in terms of statistics (non-negligible transition probabilities) that can be estimated with small relative variance. Assumption 3 ensures that other “small” entries in the matrix do not spoil the accuracy of the estimator.

---

> > ### Comment · Reviewer_251G · 2024-08-12
> >
> > I thank the author for patiently answering all my queries. However, I still have some questions:
> > 1. Confusion regarding $u(i)$: Can a rare event be identified using $u(i)$ or is the claim restricted to better estimation of $u(i)$ even in the presence of rare events?
> > 2. In your response to my earlier Question 8, you say that "TD does require escape events in its data set.". However, if there are no such events in the dataset, how will the TD estimator learn the values correctly?
> > 3. While I understand that this paper is restricted to the finite MDP setting, do the authors have any thoughts on how their analyses and conclusions hold true in the function approximation based setting with large and possibly continuous state and action spaces?
> > 4. Sorry for asking this again, but I have a slight confusion here. In the examples in Section 3, what does escape from Set D imply? What is set D in each of these examples and where does the trajectory that escapes from D go?
> >
> > I am satisfied with the other responses and accordingly am increasing my score.

---

> ### Author Response · Authors · 2024-08-12
>
> We thank you again for the opportunity to respond to your questions.
>
> 1. Our claim is that TD can be much more efficient than MC for problems involving rare events. The values of $u$ alone do not reveal whether the estimation problem is difficult. Imagine a process $X_t$ that always exits $D$ by the same sequence of steps ($X_t$ is deterministic). In this case the accumulated reward will not be random and we can tune the values of $u$ by our choice of $R$. Yet the MC estimator will have zero variance. So the distinguishing characteristic of the rare event setting is the difficulty of estimating $u$, and not just the values of $u$. The notion of relative accuracy may be contributing to confusion here. The size of $u(i)$ does affect how much variation in the estimate of $u(i)$ we are willing to accept. We need that variation to be small compared to the value of $u(i)$.
> 2. As stated in our response to Question 8, “TD **does** require escape events in its data set” and, in our numerical examples, a small fraction of trajectories **do** escape before time $\tau$. If the data set does not include escape events, the linear system defined in Eq. (2) will be indefinite and the TD estimator will be undefined. Our Assumption 1 and the large data limit ($M \to \infty$) in our theorems guarantee the existence of escapes.
> 3. The large statistical error of the MC estimator in the finite state setting translates immediately to continuous spaces, and implies large estimation errors for standard ML rare event prediction schemes that use data sets of full escape trajectories. The Markov state model (MSM) approximations mentioned at lines 144-153 solve continuous space problems by first partitioning state space (usually by data clustering) and then approximating the solution to the Bellman equation by a function that is constant on each partition. Subject to the assumptions of Section 4, our theory describes the estimation error of such a scheme in the rare event setting, but does not address approximation error. We mention this briefly at lines 34-42.
>
>     We comment on more elaborate generalizations to continuous state spaces in the conclusion (lines 337-341). Our proof of Theorem 1 is based on new perturbation bounds for finite dimensional linear systems which, we believe, have appropriate extensions to infinite dimensional settings. Such a result would say that rather large perturbations of the operator are possible without large errors in the solution to the linear system, as long as the perturbations are “local” (as represented in the finite setting by Assumptions 2 and 3) in some appropriate sense. Once we have an infinite dimensional perturbation bound of this type, we can begin to analyze both the approximation error and estimation error of TD approximations in continuous spaces.
>
> 4. On Line 203 we define the set $D$ as $[n] \setminus \\{1, n\\}$ for all the examples in Section 3. So $D^c$ is $\\{1,n\\}$ and a trajectory that escapes must reach either state 1 or state $n$ (the left or right boundaries). The definition of $u(i)$ in Eq. (1) only depends on the path of $X_t$ until it reaches $D^c$. It does not depend on what the process might have done after that. In our theoretical setup, every trajectory in the data set is initialized as an independent draw from some distribution $\mu$ (the uniform distribution in the numerical examples of Section 3). So our estimators (both MC and TD) also do not depend on the path of the process after reaching $D^c$.
>
>     As mentioned at lines 63-65, in some application settings it is common to analyze short trajectories corresponding to pieces of a single, much longer trajectory. In this case, after escaping from $D$ one must wait for the process to re-enter $D$ before gathering new data. Short trajectories gathered in this way will be correlated and, as we mention there, our analysis does not immediately apply. Looking beyond your question a little, inspection of our results does reveal some interesting, and practically important, implications for that setting. Examining the asymptotic variance formula in Eq. (5), we see that small values $\mu(k)$ can result in large errors. In fact, if we replaced the uniform initial distribution used in Sec. 3 by the invariant distribution given in Eq. (9) (which does not satisfy Assumption 1), then TD would have exponentially large relative variance. But a single very long trajectory would result in points drawn approximately from this invariant distribution. The implication is that a practitioner either needs to generate data using a different strategy (as has been emphasized in several references cited at lines 31 to 33), or, if that is not possible, reweight the data. This topic is important and we hope to focus on it more in follow up work.

---

> > ### Comment · Reviewer_251G · 2024-08-13
> >
> > Thanks for these clarifications. Based on these I am increasing my score.

---

### Official Review · Reviewer_4mMw · 2024-07-16

**Soundness:** 3
**Presentation:** 2
**Contribution:** 3
**Rating:** 6
**Confidence:** 2

**Summary:**

This paper provides a rigorous sample complexity comparison between MC sampling and LSTD method, and shows that for rate event, LSTD can provide relatively accurate estimation with a much smaller dataset compared to MC sampling. This contradicts to the intution from the classical worst perturbation bounds.

**Strengths:**

- An upper bound on the relative asymptotic variance is provided for LSTD

- This paper considers the policy evaluation for rare event, which is new to the best of my knowledge

**Weaknesses:**

- There is no simulation results for rate event scenario

- The writing can be more polished. Some explanation can be more formal. For example, it would be great if the authors can formalize their discussion in section 1.2.

**Questions:**

NA

---

> ### Author Rebuttal · Authors · 2024-08-06
>
> Thank you for your review of the paper and supportive comments.
>
> For a detailed response regarding our numerical experiments, please see our General Response.
>
> Given space constraints, our goal in Section 1.2 was to *very* briefly summarize basic matrix perturbation and quasi-stationary limit results that motivate and position our contributions. Precise statements of our theoretical results are stated in Section 4 where the key assumptions for the rare event problem are introduced. But we appreciate that we can strike a better balance of precision and brevity in Section 1.2.

---

### Author Rebuttal · Authors · 2024-08-06

We would like to thank the reviewers for their time reviewing our work, and a number of thoughtful suggestions for how to improve the presentation of the paper. We have responded to all individual comments below, but please let us know if any other questions arise.

There is a shared point of discussion raised by reviewers about the numerical experiments, which we address here. We present numerical results in Sec. 3 before the mathematical formulation of rare events setting in Sec. 4. We apologize for any confusion arising from this arrangement. We intended to motivate the development in Sec. 4 using concrete examples. We will identify this connection more clearly at the beginning of Sec. 3.

In fact, the examples in Sec. 3 satisfy the assumptions in Sec. 4, as commented after each assumption (lines 285-286 and 292-294). Also, our numerical results in Sec. 3 precisely exemplify the key theoretical findings in Sec. 4. In Figs. 1 and 2, the empirical relative MSEs of TD (circles) align with the theoretical relative asymptotic variance (lines), and support our prediction in Theorem 2 that asymptotic relative MSE will grow slowlier than $n^3$. The relative MSE of the MC estimator increases exponentially in $n$ in the right panels of Figs. 1 and 2  (note the different y-scales in the middle and right panels). While the assumptions in Sec. 4 cover more general cases, our numerical experiments in Sec. 3 concisely demonstrate the key ideas.

---

### Decision · Program_Chairs · 2024-09-25

**Decision:**

Accept (poster)

**Comment:**

This paper presents a strong theoretical contribution to the field of policy evaluation under rare events, particularly in the context of Markov chains.
The originality of the problem, the rigor of the analysis, and the clarity of the presentation (as noted by one reviewer) make it a valuable contribution.
The authors' rebuttal effectively addresses concerns about the numerical experiments, clarifying that empirical results do support the theoretical claims and enhancing the connection between theory and practice.